# Improving Offline-to-Online Reinforcement Learning with Q Conditioned State Entropy Exploration

## Abstract

Studying how to fine-tune offline reinforcement learning (RL) pre-trained policy is profoundly significant for enhancing the sample efficiency of RL algorithms. However, directly fine-tuning pre-trained policies often results in sub-optimal performance. This is primarily due to the distribution shift between offline pre-training and online fine-tuning stages. Specifically, the distribution shift limits the acquisition of effective online samples, ultimately impacting the online fine-tuning performance. In order to narrow down the distribution shift between offline and online stages, we proposed Q conditioned state entropy (QCSE) as intrinsic reward. Specifically, QCSE maximizes the state entropy of all samples individually, considering their respective Q values. This approach encourages exploration of low-frequency samples while penalizing high-frequency ones, and implicitly achieves State Marginal Matching (SMM), thereby ensuring optimal performance, solving the asymptotic sub-optimality of constraint-based approaches. Additionally, QCSE can seamlessly integrate into various RL algorithms, enhancing online fine-tuning performance. To validate our claim, we conduct extensive experiments, and observe significant improvements with QCSE ( about **13**% for CQL and **8**% for Cal-QL). Furthermore, we extended experimental tests to other algorithms, affirming the generality of QCSE.

## 1 Introduction

Offline RL holds a unique advantage over online RL, as it can be exclusively trained using pre-existing static offline RL datasets, eliminating the needs for interactions with the environment to acquire new online samples (Levine et al., 2020). However, offline RL encounters specific limitations, including the challenges of learning sub-optimal performance, and the risks of overestimating out-of-distribution (OOD) state actions (Kumar et al., 2020a), and limited exploration, particularly when the offline dataset is sub-optimal and fails to provide sufficient coverage (Lei et al., 2024). Therefore, we need to further fine-tune the offline pre-trained policy in online setting to address the aforementioned limitations (Fujimoto and Gu, 2021; Kostrikov et al., 2021; Wu et al., 2022; Mark et al., 2023).

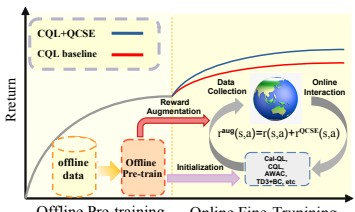

Figure 1: Demonstration of QCSE.

Drawing the inspiration from modern machine learning, where pre-training is succeeded by online fine-tuning on downstream tasks (Brown et al., 2020; Touvron et al., 2023), it seems plausible to further enhance the performance of offline pre-trained policy through the process of online fine-tuning. However, offline algorithms often confine the offline pre-trained policy's likelihood within offline support, resulting in unstable fine-turning performance due to distribution shifts between offline pre-training and online fine-tuning. To address these limitations, previous works attempt to introduce additional constrains (Kostrikov et al., 2021; Zheng et al., 2022; Zhao et al., 2023; Wu et al., 2022) during the online fine-turning stage. However, these algorithms still suffer from a demonstrated performance decline (as shown by Nakamoto et al. in Figure 1) or asymptotic sub-optimality during the initial online fine-tuning stage (Nakamoto et al., 2023; Wu et al., 2022; Li et al.,

2023). In addition to the predominant regularization-based approaches Nakamoto et al. introduce a method aimed at aligning the value estimation during offline and online stages, thereby ensuring standard online fine-turning.

From a general perspective, all of these approaches can mitigate or resolve the negative impact caused by distribution shifts between offline and online stages. Specifically, constraint-based approaches maximize action prediction within the action support and then gradually loosen the constraints, thus alleviating policy collapse caused by distribution shifts. Regarding the value-aligned approach, ensuring standard online fine-tuning involves aligning the value estimation during both offline pre-training and online fine-tuning stages, which mitigate the distribution shift in value estimation between offline and online stages, consequently ensuring stable and asymptotic policy fine-tuning performance. Therefore, such insight opens a question that: *Can we focus on exploring effective samples to alter the distribution shift between offline and online stages to guarantee the online fine-tuning performance?*

To answer this question, we showcase the necessity of state entropy maximization in online fine-tuning stage: *1) Exploring diverse online samples help to diminish the negative impact of conservative pre-training.* Specifically, it's feasible to gather diverse samples to mitigate the conservatism of offline pre-trained policy (Luo et al., 2023) and further collect effective online samples to improve the fine-tuning performance. *2) Effective exploration is the crucial factor in enabling the asymptotically optimality.* In particular, if the exploration can ensure that the marginal state distribution matches the target (expert) density *i.e.* State Marginal Matching (SMM) (Lee et al., 2020), it can further guarantee that the empirical policy approaches the optimal policy.

Therefore, it's intuitively plausible to reduce the distribution shift between the offline and online learning stages by gathering diverse samples, while approximately reaching the optimal performance via theoretically optimal exploration, such as State Marginal Matching (SMM). Based on this insight, we introduced Q conditioned State Entropy (QCSE), which involves separately estimating the state entropy conditioned on the Q estimates of each sample. By maximizing their average, we implicitly achieve SMM, aiding the empirical policy approaching the optimal policy. The advantage of QCSE lies in that it can approximately realize SMM, and thus naturally solve the asymptotic sub-optimality of constraint-based approaches. Meanwhile, QCSE inherently benefits from the exploratory advantages of value conditioned reward designing represented by Kim et al.. Specifically, Q-conditioned state entropy maximization can alleviate biased exploration towards low-value state regions as the state distribution becomes more uniform. In particular QCSE is different from previous value conditioned reward designing where QCSE resolves the biased exploration of value condition by considering differences in transitions that were previously overlooked, thus improving the online fine-tuning stability. Additionally, QCSE holds theoretical advantage and innovation in offline-to-online setting, while value conditioned reward design only considers the online setting. *To summarize, the contribution of our research can be summerized as follows:*

- We will imply that state entropy maximization can implicitly realize SMM in online stage, and further contribute to approaching the optimal performance.

- Based on the theoretical advantage of SMM, we proposed Q conditioned state entropy exploration (QCSE) that can implicitly realize SMM during online fine-tuning stage.

- Compared with VCSE, QCSE takes transition information into consideration, thereby protecting transitions from being disrupted by indiscriminate entropy maximization.

## 2 RELATED WORK

**Offline RL.** The notorious challenges in offline RL pertains to the mitigation of out-of-distribution (OOD) issues, which are a consequence of the distributional shift between the behavior policy and the training policy (Fujimoto et al., 2019a). To effectively address this issue, **1)** conservative policy-based *model-free* methods adopt the following approaches: Adding policy regularization (Fujimoto et al., 2019b; Kumar et al., 2019; Wu et al., 2019; Liu et al., 2023), or implicit policy constraints (Peng et al., 2019; Siegel et al., 2020; Zhou et al., 2020; Chen et al., 2022; Wu et al., 2022; Liu et al., 2023). **2)** And, conservative critic-based model-free methods penalize the value estimation of OOD state-actions via conducting pessimistic Q function (Kumar et al., 2020a) or uncertainty estimation (An et al., 2021; Bai et al., 2022; Rezaeifar et al., 2022; Wu et al., 2021) or implicitly

regularizing the bellman equation (Kumar et al., 2020b). In terms of the *model-base* offline RL, it similarly train agent with distribution regularization (Hishinuma and Senda, 2021), uncertainty estimation (Yu et al., 2020; Kidambi et al., 2020; Lu et al., 2022), and value conservation (Yu et al., 2021). In our research, due to the remarkable sampling efficiency and outstanding performance of model-free algorithms in both offline and online RL settings, and we prove that QCSE satisfy the guarantee of Soft-Q optimization (theorem 4.1), thus we select *Conservative Q-Learning* (CQL) and *Calibrated Q-Learning* (Cal-QL) as our primary baseline methods. Additionally, to conduct a thorough assessment of the effectiveness of our proposed approaches, we have also expanded our evaluation to encompass a diverse set of other model-free algorithms, including *Soft-Actor-Critic* (SAC) (Haarnoja et al., 2018), *Implicit Q-learning* (IQL) (Kostrikov et al., 2021), *TD3+BC* (Fujimoto and Gu, 2021), and *AWAC* (Nair et al., 2021).

**Offline-to-Online RL.** Previous researches have demonstrated that offline RL methods offer the potential to expedite online training, a process that involves incorporating offline datasets into online replay buffers (Nair et al., 2021; Vecerik et al., 2018; Todd Hester and et al., 2017) or initializing the pre-trained policy to conduct online fine-tuning (Kostrikov et al., 2021; Beeson and Montana, 2022). However, there exhibits worse performance when directly fine-tuning the offline pre-trained policy (Nakamoto et al., 2023; Lee et al., 2021a), and such an issue can be solved by adapting a balanced replay scheme aggregated with pessimistic pre-training (Lee et al., 2021a), or pre-training with pessimistic Q function and fine-tuning with exploratory methods (Wu et al., 2022; Mark et al., 2023; Nakamoto et al., 2023). In particular, our approach QCSE differs from these methods in that it enhances online fine-tuning solely by augmenting online exploration. (More related works have been added to Appendix G.1)

**Online Exploration.** Recent advances in the studies of exploration can obviously improve online RL sample efficiency, among that, remarkable researches include injecting noise into state actions(Lillicrap et al., 2019) or designing intrinsic reward by counting visitation or errors from predictive models (Badia et al., 2020; Sekar et al., 2020; Whitney et al., 2021; Burda et al., 2018). In particular, the approaches most related to our study are to utilize state entropy as an intrinsic reward (Kim et al., 2023; Seo et al., 2021).

## 3   PRELIMINARY

**Reinforcement Learning (RL).** We formulate RL as Markov Decision Process (MDP) tuple *i.e.* $\mathcal{M} = (\mathcal{S}, \mathcal{A}, r, d_{\mathcal{M}}, p_0, \gamma)$. Specifically, $p_0$ denotes the distribution of initial state (observation), $\mathbf{s}_0 \sim p_0$ denotes initial observation, $\mathcal{S}$ denotes the observation space, $\mathcal{A}$ denotes the actions space, $r(\mathbf{s}, \mathbf{a}) : \mathcal{S} \times \mathcal{A} \mapsto \mathbb{R}$ denotes the reward function, $d_{\mathcal{M}}(\mathbf{s}'|\mathbf{s}, \mathbf{a}) : \mathcal{S} \times \mathcal{A} \to \mathcal{S}$ denotes the transition function (dynamics), and $\gamma \in [0, 1]$ denotes discount factor. The goal of RL is to obtain the optimal policy $\pi^* : \mathcal{S} \mapsto \mathcal{A}$ to maximize the accumulated discounted return *i.e.* $\pi^* := \arg\max_\pi \mathbb{E}_{\tau \sim \pi}[R(\tau)]$, where $\mathbb{E}_{\tau \sim \pi}[R(\tau)] = \mathbb{E}[\sum_{t=0}^{t=T} \gamma^t r(\mathbf{s}_t, \mathbf{a}_t)]$, and $\tau = \{\mathbf{s}_0, \mathbf{a}_0, r_0, \cdots, \mathbf{s}_T, \mathbf{a}_T, r_T | \mathbf{s}_0 \sim p_0, \mathbf{a}_t \sim \pi(\cdot|\mathbf{s}_t), \mathbf{s}_{t+1} \sim d_{\mathcal{M}}(\cdot|\mathbf{s}_t, \mathbf{a}_t)\}$ is the rollout trajectory. Generally, RL has to estimate a Q function *i.e.* $Q(\mathbf{s}, \mathbf{a}) = \mathbb{E}_{\tau \sim \pi}[\sum_{t=0}^{T} \gamma^t r(\mathbf{s}_t, \mathbf{a}_t) | \mathbf{s}_0 = \mathbf{s}, \mathbf{a}_0 = \mathbf{a}]$, and a value function by $V(\mathbf{s}) := \mathbb{E}_{\mathbf{a} \sim \mathcal{A}}[Q(\mathbf{s}, \mathbf{a})]$ to assistant in updating policy $\pi$. In this research, we mainly focus on improving model-free algorithms to conduct offline-to-online RL setting.

**Model-free Offline RL.** Typically, model-free RL algorithms alternately optimize policy with Q-network i.e. $\pi := \arg\max_\pi \mathbb{E}_{\mathbf{s} \sim \mathcal{D}, \mathbf{a} \sim \pi(\cdot|\mathbf{s})}[Q(\mathbf{s}, \mathbf{a})]$, and conduct policy evaluation by the Bellman equation iteration i.e. $Q := \arg\min_Q \mathbb{E}_{(\mathbf{s}, \mathbf{a}, \mathbf{s}') \sim \mathcal{D}}[(Q(\mathbf{s}, \mathbf{a}) - \mathcal{B}_{\mathcal{M}} Q(\mathbf{s}, \mathbf{a}))^2]$, where $\mathcal{B}_{\mathcal{M}} Q(\mathbf{s}, \mathbf{a}) := r(\mathbf{s}, \mathbf{a}) + \gamma \cdot Q(\mathbf{s}', \pi(\cdot|\mathbf{s}'))$. In particular, model-free offline RL aims to learn from the static RL datasets $\mathcal{D}$ collected by behavior policy $\pi_\beta$ without access to the environment to collect new trajectories, thus suffers from the OOD state actions and sub-optimal performance, therefore, it's necessary to further fine-tune the pre-trained policy online to further alleviate the limitations of offline pre-trained policy.

**Drawbacks of previous offline-to-online algorithms.** However, directly fine-tuning the offline pretrained policy may encounter distribution shift issues, potentially leading to policy collapse. Despite that constraint-based approaches can facilitate stable online fine-tuning, they may suffer from

sub-asymptotic optimality. Different from the majority of offline-to-online approaches, we propose Q conditioned state entropy exploration (QCSE), which alleviates distribution shift issue by implicitly realizing SMM through Q conditioned state entropy maximization.

## 4 Q CONDITIONED STATE ENTROPY MAXIMIZATION (QCSE)

Previously, Lee et al. suggests that error or count based exploration approaches (Pathak et al., 2017; Burda et al., 2018) are insufficient to converge to a singular exploratory policy, and propose SMM to realize converging to singular exploratory policy. Building on the theoretical advantages of SMM, we aim to enhance online exploration through state entropy maximization, thereby improving online fine-tuning performance.

**Definition 1** (Marginal State distribution). *Given the rollout trajectory $\tau \sim \pi$, we define the marginal state distribution of $\pi$ as $\rho_\pi(\mathbf{s}) = \mathbb{E}_{\mathbf{s}_0 \sim p_0, \mathbf{a}_t \sim \pi(\cdot|\mathbf{s}_t), \mathbf{s}_{t+1} \sim d_\mathcal{M}(\cdot|\mathbf{s}_t, \mathbf{a}_t)}[\frac{1}{N} \sum_{t=1}^{T} 1(\mathbf{s}_t = \mathbf{s})]$.*

**Definition 2** (State Marginal Matching). *Given the target (optimal) state density $p^*(\mathbf{s})$ and the offline initialized marginal state distribution $\rho_\pi(s)$. We define State Marginal Matching (SMM) as optimizing policy to minimize $D_{\mathrm{KL}}[\rho_\pi(\mathbf{s})||p^*(\mathbf{s})]$, i.e. $\pi := \arg\min_\pi D_{\mathrm{KL}}[\rho_\pi(\mathbf{s})||p^*(\mathbf{s})]$, where $D_{\mathrm{KL}}$ denotes Kullback-Leibler (KL) divergence* [1].

**Definition 3** (Critic Conditioned State Entropy). *Given empirical policy $\pi \in \Pi$, its critic network $Q(\mathbf{s}, \mathbf{a}) : \mathcal{S} \times \mathcal{A} \to \mathbb{R}$, and given state density of current empirical policy: $\rho_\pi(\mathbf{s})$, where $\int_{\mathbf{s} \in \mathcal{D}} \rho_\pi(\mathbf{s}) = 1$. We define the critic conditioned entropy as $\mathcal{H}_\pi(\mathbf{s}|Q) = \mathbb{E}_{\mathbf{s} \sim \rho_\pi}[-\log p(\mathbf{s}|Q(\mathbf{s}, \pi(\cdot|\mathbf{s})))]$.*

To explain why state entropy maximization helps to address the challenges of offline-to-online RL, we first define the basic concepts and notations, specifically, we define the marginal state distribution as Definition 1, and the process of the marginal state distribution $\rho_\pi(\mathbf{s})$ approaching target density $p^*(\mathbf{s})$ *i.e.* state marginal matching (SMM) as Definition 2. Subsequently, we illustrate how maximizing state entropy can approximate SMM during the online fine-tuning stage, thereby facilitating the acquisition of the optimal policy.

**State entropy maximization implicitly realize SMM during the online fine-tuning stage.** To elaborate on the relationship between state entropy maximization and SMM, we derive the process of maximizing the state entropy of the empirical policy $\pi$ *i.e.* $\max \mathbb{E}_{\mathbf{s} \sim \rho_\pi}[\mathcal{H}_\pi[\mathbf{s}]] := \max \mathbb{E}_{\mathbf{s} \sim \rho_\pi}[-\log p(\mathbf{s})], s.t.\pi := \arg\max_\pi Soft\text{-}Q^\pi$, and arrive at the following expression:

$$\max_{\substack{\rho_\pi \\ s.t. \max_\pi SoftQ^\pi}} \mathbb{E}_{\mathbf{s} \sim \rho_\pi}[\mathcal{H}_\pi[\mathbf{s}]] \leq \max_{\substack{\rho_\pi \\ s.t. \max_\pi SoftQ^\pi}} \underbrace{\int_{\mathbf{s} \sim \mathcal{S}_2} -\rho_\pi(\mathbf{s}) \log \rho_\pi(\mathbf{s})}_{J_{\mathrm{term1}}} + \underbrace{\int_{\mathbf{s} \sim \mathcal{S}_1} -p^*(\mathbf{s}) \log p^*(\mathbf{s})}_{J_{\mathrm{term2}}}, \quad (1)$$

where $p^*(\mathbf{s})$ denotes the target density, $\rho_\pi$ denotes the marginal state distribution initialized by the offline dataset, as defined in Definition 1. $\mathcal{S}_1$ denotes domain where $\rho_\pi(\mathbf{s}) > p^*(\mathbf{s})|_{\mathbf{s} \sim \mathcal{S}_1}$, and $\mathcal{S}_2$ denotes domain where $\rho_\pi(\mathbf{s}) \leq p^*(\mathbf{s})|_{\mathbf{s} \sim \mathcal{S}_2}$. *proof* of Equation 1 see Appendix B.1.

Since $p^*(s)$ remains invariant during the training process, maximizing $J_{\mathrm{term2}}$ is equivalent to narrowing down the domain $S_1$. Meanwhile, maximizing $J_{\mathrm{term1}}$ is equivalent to encourage exploring $S_2$. Both $J_{\mathrm{term_1}}$ and $J_{\mathrm{term_2}}$ narrow the gap between $\rho_\pi(s)$ and $p^*(s)$. Therefore, state entropy maximization approximately solves the distribution shift issue and facilitates the approximation of SMM during online fine-tuning stage. Furthermore, it encourages the empirical policy approaching the optimal policy. Based on this analysis, we introduce QCSE that computes Q conditioned entropy as intrinsic reward to narrow down the distribution shift between offline and online stages.

**Connection with un-biased SMM (Lee et al., 2020).** In particular, we emphasize that Equation 1 is an inequality, where our goal is to maximize the left-hand side, but we can also influence its upper bound, which is the right-hand side of the inequality. Additionally, this paper does not implement SMM in an unbiased manner, but as we have discussed above, maximizing the right-hand side of

---

[1] For the given distributions $p(x)$ and $p(x)$ on the domain $X$. KL divergence denotes $D_{\mathrm{KL}}[p(x)||q(x)] = \mathbb{E}_{x \sim X}[p(x) \log \frac{p(x)}{q(x)}]$.

the equation serves the same purpose as SMM, which is to bring the current state distribution closer to the optimal distribution. Furthermore, our problem setting is a RL problem, and when optimizing Equation 1, the RL objective can influence the state distribution to explore high-value samples as much as possible, thereby promoting the convergence of the distribution to the optimal distribution.

**Implementation of QCSE.** The mathematical formulation of QCSE, as shown in Equation 2, involves calculating the Q conditioned state entropy as intrinsic reward to encourage the agent to explore the environment uniformly. In particular, we use $\mathrm{Tanh}$ to limit the magnitude of the $r^{\mathrm{QCSE}}$, preventing it from overwhelming the task reward and thereby reducing biases during critic updating.

$$r^{\mathrm{mod}}(\mathbf{s}, \mathbf{a}) = \lambda \cdot \mathrm{Tanh}(\underbrace{\mathcal{H}(\mathbf{s}|\min(Q_{\phi_1}(\mathbf{s}, \mathbf{a}), Q_{\phi_2}(\mathbf{s}, \mathbf{a})))}_{r^{\mathrm{QCSE}}}) + r(\mathbf{s}, \mathbf{a})\bigg|_{(\mathbf{s}, \mathbf{a}) \sim \mathcal{D}_{\mathrm{online}}}, \qquad (2)$$

where $\phi_1$ and $\phi_2$ are separately the params of double Q Networks. However, we cannot directly obtain $\rho_\pi(\mathbf{s})$, therefore, we cannot directly calculate state entropy. In order to approximate $\rho_\pi(\mathbf{s})$, we refer to Kim et al. (2023), and use the Kraskov-Stögbauer-Grassberger (KSG) estimator (Kraskov et al., 2003) to approximate state entropy as the intrinsic reward, *i.e.* Equation 3.

$$r^{\mathrm{QCSE}}(\mathbf{s}, \mathbf{a}) = \frac{1}{d_s}\phi(n_{\hat{Q}_\phi}(i) + 1) + \log 2 \cdot \max(||\mathbf{s}_i - \mathbf{s}_i^{knn}||, ||\hat{Q}(\mathbf{s}, \mathbf{a}) - \hat{Q}(\mathbf{s}, \mathbf{a})^{knn}||)\bigg|_{(\mathbf{s}, \mathbf{a}) \sim \mathcal{D}_{\mathrm{online}}}, \tag{3}$$

where $\hat{Q}(\mathbf{s}, \mathbf{a}) = \min(Q_{\phi_1}(\mathbf{s}, \mathbf{a}), Q_{\phi_2}(\mathbf{s}, \mathbf{a}))$, and $\mathbf{s}_i^{knn}$ is the $n_\mathbf{s}(i)$-th nearest neighbor of $\mathbf{s}_i$ in the space $\mathcal{S}$ spanned by $\mathbf{s}_i$. Specifically, $n_\mathbf{s}(i)$ represents the number of neighborhoods around $\mathbf{s}_i$, where the distances to $\mathbf{s}_i$ are less than $\frac{\epsilon(\mathbf{s}_i)}{2}$. And $\epsilon(\mathbf{s}_i)$ represents twice the distance from $\mathbf{s}_i$. $n_{\hat{Q}}(i)$ and $\hat{Q}_i^{knn}$ can be similar defined by replacing $\mathbf{s}$ by $\hat{Q}$. For more detail information, please refer to page 4, line 8 to 9 of Kim et al. (2023)).

## 4.1 ADVANTAGES OF QCSE

**Monotonic of QCSE.** Despite the concise and simple mathematical form of QCSE, QCSE has advantage lies in guaranteeing the monotonic Soft-Q optimization (supported by theorem 4.1), thus can be seamlessly utilized in conjugation with Soft-Q based RL algorithms including CQL-SAC (Kumar et al., 2020a) etc.

**Theorem 4.1** (Converged QCSE Soft Policy is Optimal). *Repetitive using lemma B.1 and lemma B.2 to any $\pi \in \Pi$ leads to convergence towards a policy $\pi^*$. And it can be proved that $Q^{\pi^*}(\mathbf{s}_t, \mathbf{a}_t) \geq Q^\pi(\mathbf{s}_t, \mathbf{a}_t)$ for all policies $\pi \in \Pi$ and all state-action pairs $(\mathbf{s}_t, \mathbf{a}_t) \in \mathcal{S} \times \mathcal{A}$, provided that $|\mathcal{A}| < \infty$.* ***Proof*** *see Appendix B.3.*

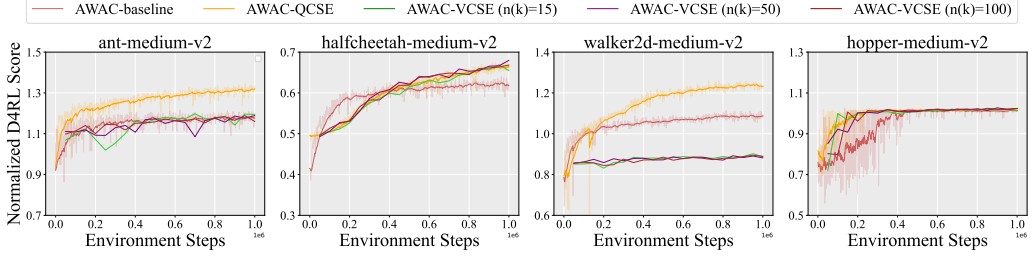

Figure 2: Q condition vs. V condition. In this experiment, we selected AWAC as the base algorithm and compared using V network and Q network to calculate the intrinsic reward's condition. The experimental results indicate that using the Q-network to compute the condition leads to overall better performance for AWAC. Nair et al. (2021) points out that AWAC demonstrates poor online fine-tuning performance.

**Q condition protects transitions from being disrupted by entropy maximization.** Another advantage of QCSE is that it utilizes Q instead of V as the intrinsic reward's condition. This approach incorporates transition information, thereby preserving transitions that disrupted by indiscriminate

state entropy maximization. For instance, assuming that there exists two transitions $T_1 = (\mathbf{s}, \mathbf{a}_1, \mathbf{s}_1)$ and $T_2 = (\mathbf{s}, \mathbf{a}_2, \mathbf{s}_2)$. Since $T_1$ and $T_2$ have the same current observation $\mathbf{s}$, they are related to the same value conditioned intrinsic reward $-\log(\mathbf{s}|V(\mathbf{s}))$, therefore, low-value transitions still receive relatively high intrinsic rewards, subsequently biasing the Q estimation, and further impacting decision-making. However, by computing intrinsic rewards conditioned on $Q(\mathbf{s}, \mathbf{a})$ and maximizing state entropy with distinct Q values separately, QCSE can further mitigate the biased exploration issues left unresolved by VCSE. To validate our claims, we chose AWAC as baseline, and separately utilize both Q-network and V-network to compute the intrinsic reward's condition, conducting tests on tasks sourced from Gym-Mujoco domain. As shown in Figure 2, using Q to compute condition has better performance compared to using V.

## 5 PRACTICAL IMPLEMENTATION

We follow the standard offline-to-online RL process to test QCSE. Specifically, we first pre-train the policy with the selected algorithm on a specific offline dataset. Then, we further fine-tune the pre-trained policy online using QCSE. Finally, we test using the policy fine-tuned online. In terms of the real implementation, QCSE augments the reward of online dataset by calculating the Q conditional state entropy (via Equation 3) which is highly compatible with Q-ensemble or double-Q RL algorithms. For algorithms that do not employ Q-ensemble or double Q, it is still possible to use QCSE; however, they may not benefit from the advantages associated with Q-ensemble, as clarified in the following section. When it comes to the hyper-parameters of QCSE, setting $\lambda$ in Equation 2 to 1 is generally sufficient to improve the performance of various baselines on most tasks. However, it is important to note that QCSE's effectiveness is influenced by the number of k-nearest neighbor (knn) clusters, as we have demonstrated in our ablation study. Additionally, for parameters unrelated to QCSE, such as those of other algorithms used in conjunction with QCSE, it is not necessary to adjust the original parameters of these algorithms (see more details including **hyper-parameters**, **model architecture**, **compiting recourese**, e.g. in Appendix D.6). In the following section we will conduct experimental evaluation to validate the effectiveness of QCSE.

## 6 EVALUATION

Our experimental evaluation aims to achieve the following primary objectives: **1)** Investigating the effectiveness of QCSE utilizing soft-Q based algorithms in facilitating offline-to-online RL and assessing its performance. **2)** Assessing the viability of integrating QCSE with another model-free algorithms to enhance their sample efficiency. **3)** Conducting diverse experiments to showcase the performance differences or relationships between QCSE and various exploration methods, including SE (Seo et al., 2021), VCSE (Kim et al., 2023), RND (Burda et al., 2018), and SAC (Haarnoja et al., 2018). **4)** Conducting a series of ablation experiments to verify the effectiveness of QCSE. We first introduce our tasks and baselines.

**Task and Datasets.** We experiment with 12 tasks from mujoco (Brockman et al., 2016) and Antmaze in D4RL (Fu et al., 2021). Meanwhile, we also test QCSE on tasks from binary androit domain (see Nair et al.). The selected tasks cover various aspects of RL challenges, including reward delay and high-dimensional continuous control. Specifically: **1)** In the Antmaze tasks, the goal is to control a quadruped robot to reach the final goal. Notably, this agent does not receive an immediate reward for its current decision but instead only receives a reward of +1 upon successfully reaching the goal or terminating. This setup presents a form of reward delay, making these tasks adapt to evaluate the long horizontal decision-making capability of algorithms. **2)** In Gym-locomotion tasks, the goal is to enhance the agent's localmotion capabilities, presenting a contrast to the Antmaze domain where Gym-Mujoco tasks feature high-dimensional decision-making spaces. Also, the agent in Gym-Mujoco has the potential to obtain rewards in real time. Additionally, we use `binary-Androit` tasks to assess the performance of QCSE on Androit-related tasks (For much more details about this domain please refer to Nair et al.; Nakamoto et al.).

**Baselines for Comparison.** For convenience, we name any algorithm **Alg** paired with QCSE as **Alg**-QCSE. Now we introduce our baselines. We primarily compare CQL-QCSE and Cal-QL-QCSE to **CQL** (Kumar et al., 2020a) and **Cal-QL** (Nakamoto et al., 2023). We also verify that QCSE can be broadly plugged into various model-free algorithms including **SAC** (Haarnoja et al., 2018),

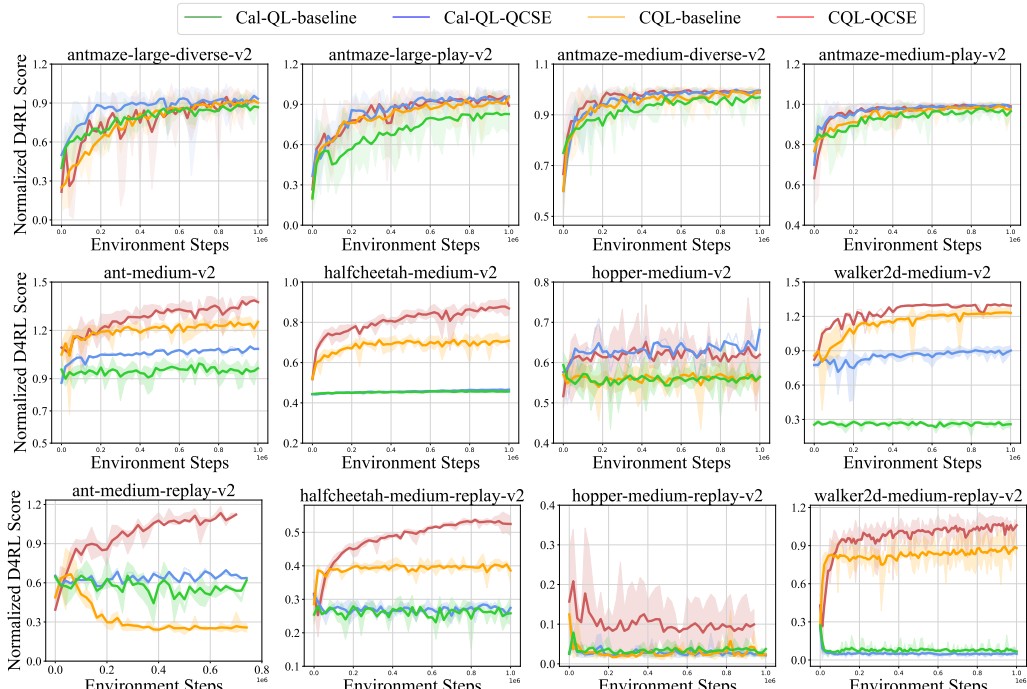

Figure 3: Online fine-tuning curve on selected tasks. We tested QCSE by comparing Cal-QL-QCSE, CQL-QCSE to Cal-QL, CQL on selected tasks in the Gym-Mujoco and Antmaze domains, and then reported the average return curves of multi-time evaluation. As shown in this Figure, QCSE can improve Cal-QL and CQL's offline fine-tuning sample efficiency and achieves better performance than baseline (CQL and Cal-QL *without* QCSE) *over all selected tasks*.

**IQL** (Kostrikov et al., 2021), **TD3+BC** (Fujimoto and Gu, 2021), and **AWAC** (Nair et al., 2021), thus improving their online fine-tuning performance. In particular, Cal-QL is the recent state-of-the-art (SOTA) offline-to-online RL algorithm that has been adequately compared to multiple offline-to-online methods (O3F (Mark et al., 2023), ODT (Zheng et al., 2022), and mentioned baselines), and demonstrated obvious advantages. In particular, we explain the motivation we choose these baselines. Specifically, Figure 3 uses CQLCal-QL to validate our proposed theory (theorem 4.1). Figure 4 employs AWAC, TD3-BC and IQL to demonstrate the algorithm's generalizability. Table 3 selects CQL and CalQL to verify the effectiveness of exploration algorithms in offline-to-online.

## 6.1 MAIN RESULTS

**Can QCSE improve offline-to-online RL?** It is clear that QCSE enhances the online fine-tuning performance of both CQL (improve 13%) and Cal-QL (improve 8%). This is evidenced by the stable and progressed fine-tuning curves shown in Figure 3 and the fine-tuned results shown in Table 3. Specifically, CQL and Cal-QL combined with QCSE exhibit fewer gaps in the training curves, indicating that QCSE helps to reduce the domain shift between the offline and online stages, thereby benefiting the online fine-tuning process. Additionally, CQL-QCSE outperforms both CQL and Cal-QL on nearly all selected tasks, supporting our hypothesis that enhancing the agent's exploration capabilities by state entropy maximization can help guarantee the asymptotic optimality of the online fine-tuned policy via implicitly achieving State Marginal Matching (SMM). Furthermore, when examining tasks with larger distribution shifts, such as `medium-replay` and `medium`, CQL-QCSE demonstrates better asymptotic optimality than CQL, reinforcing this claim. (In Table 3, we present the average training results from the last 20 to 100 steps across multiple runs for `Antmaze` and `gym-medium`. For `gym-medium-replay`, we report the average of the maximum values obtained from multiple runs.) Meanwhile, we conducted tests on the binary adroit domain. As shown in Table 4, QCSE consistently enhances the performance of both CQL and Cal-QL in the binary adroit domain.

Table 1: Normalized score after online fine-tuning. We report the online fine-tuned normalized return. QCSE obviously improves the performance of CQL and Cal-QL. In particular, CQL-QCSE (mean score of **92.5**) is the best out of the 12 selected baselines. Notably, part of Antmaze's baseline results are *quoted* from existing studies. Among them, AWAC's results are *quoted* from Kostrikov et al. (2021) and CQL's results are *quoted* from Nakamoto et al. (2023). Addtionally we have report the

| Offline-to-online Tasks | IQL | AWAC | TD3+BC | CQL | CQL+QCSE | Cal-QL | Cal-QL+QCSE |
|---|---|---|---|---|---|---|---|
| antmaze-large-diverse | 59 | 00 | 00 | 89.2±3.2 | 89.8±3.2 | 86.3±0.2 | **94.5±1.7** |
| antmaze-large-play | 51 | 00 | 00 | 91.7±3.8 | 92.6± 1.3 | 83.3±9.0 | **95.0±1.1** |
| antmaze-medium-diverse | 92 | 00 | 00 | 89.6±0.3 | 98.9±0.2 | 96.8±1.0 | **99.6±0.1** |
| antmaze-medium-play | 94 | 00 | 00 | 97.7±0.2 | **99.4±0.4** | 95.8±0.9 | 98.9±0.6 |
| *Total (Antmaze)* | 296 | 00 | 00 | 368.2 | 380.7 | 352.2 | **388.0** |
| halfcheetah-medium | 57 | 67 | 49 | 69.9±1.0 | **87.9±2.3** | 45.6±0.0 | 46.9±0.0 |
| walker2d-meidum | 93 | 91 | 82 | 123.1±4.0 | **130.0±0.0** | 80.3±0.4 | 90.0±3.6 |
| hopper-medium | 67 | 101 | 55 | 56.4±0.4 | **62.4± 1.3** | 55.8±0.7 | 61.7±2.6 |
| ant-medium | 113 | 121 | 43 | 123.8±1.5 | **136.9±1.6** | 96.4±0.3 | 104.2±3.0 |
| halfcheetah-medium-replay | 54 | 44 | 49 | 42.0±1.9 | **55.6±0.5** | 32.6±0.6 | 32.9±1.7 |
| walker2d-medium-replay | 90 | 73 | 90 | 98.1±5.7 | **112.7±1.5** | 27.2±8.7 | 47.7±6.4 |
| hopper-medium-replay | **91** | 56 | 88 | 17.6±11.1 | 27.1±15.8 | 13.8±2.1 | 7.7±4.1 |
| ant-medium-replay | 123 | **127** | 127 | 84.7±3.6 | **116.6±3** | 83.1±0.7 | 73.1±8.3 |
| *Total (Gym Mujoco)* | 688 | 680 | 583 | 615.6 | **729.2** | 434.8 | 464.2 |
| **Avg (Antmaze&Gym Mujoco)** | 82.2 | 56.7 | 48.6 | 82.0 | 92.5 | 65.6 | 71.0 |

Table 2: The experimental results of QCSE on the binary Androit tasks. For further details on the binary Androit task, please refer to the AWAC or CalQL.

| Algorithms | door-binary-v0 | pen-binary-v0 | Total |
|---|---|---|---|
| CQL | 99±9.9 | 20±6.6 | 119 |
| CQL+QCSE | **100**±0.0 | 95±21.8 | 195 |
| CalQL | 98±14.0 | 97±17.05 | 195 |
| CalQL+QCSE | 99±9.9 | **99**±9.7 | **198** |

**Can QCSE be plugged into other model-free algorithms?** To address the second question, we conducted comparative experiments, assessing the performance of our QCSE across various model-free algorithms such as TD3+BC, AWAC, IQL, and SAC. Notably, our QCSE, functioning as a

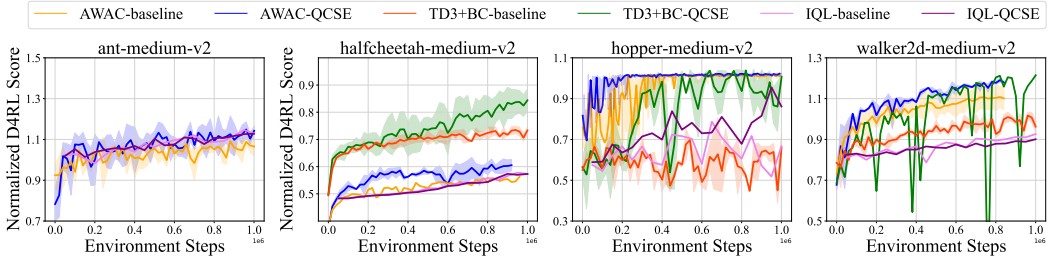

Figure 4: Performance of **Alg**-QCSE. We test QCSE with AWAC, TD3+BC, and IQL on selected Gym-Mujoco tasks, QCSE can obviously improve the performance of these algorithms on selected Gym-Mujoco tasks, showing QCSE's versatility.

plug-and-play reward augmentation algorithm, eliminates the need for additional modifications (*i.e.* **we seamlessly integrate QCSE to modify the reward during training with those algorithms**). Illustrated in Figure 4, the incorporation of QCSE leads to performance improvements across almost all algorithms during online fine-tuning. This demonstrates the versatile applicability of QCSE to a wide range of RL algorithms, extending beyond the scope of CQL or Cal-QL.

**Is QCSE more effective than previous exploration methods in the offline-to-online setting?** QCSE increases exploration to improve the performance of offline-to-online. However, there are numerous exploration-enhancing algorithms available, such as RND, VCSE, and others. It is challenging to determine whether such exploration-enhancing methods can similarly improve offline-to-

Table 3: Comparison of QCSE, VCSE and RND (Burda et al., 2018). We test CalQL and CQL algorithms, integrating them with QCSE, VCSE and RND across selected tasks in the Gym and Antmaze domains, and record the results following online fine-tuning.

| Offline-to-online Tasks | CQL | CQL+QCSE | CQL+VCSE | CQL+RND | Cal-QL | Cal-QL+QCSE | Cal-QL+VCSE | Cal-QL+RND |
|---|---|---|---|---|---|---|---|---|
| antmaze-large-diverse | 89.2±3.2 | 89.8±3.2 | 80.9±10.5 | 89.5 ±6.5 | 86.3±0.2 | 94.5±1.7 | 93.5±5.0 | 81.5±4.5 |
| antmaze-large-play | 91.7±3.8 | 92.6±1.3 | 92.2±4.9 | 92.0±6.6 | 83.3±9.0 | 95.0±1.1 | 89.5±7.6 | 84.0±7.3 |
| *Total (Antmaze)* | 180.9 | 182.3 | 173.1 | 181.5 | 169.6 | 189.5 | 183.0 | 165.5 |
| halfcheetah-medium | 69.9±1.0 | 87.9±2.3 | 64.5± 1.5 | 64.4±0.9 | 45.6±0.0 | 46.9±0.0 | 42.6± 0.1 | 41.9±0.8 |
| walker2d-meidum | 123.1±4.0 | 130.0±0.0 | 101.1±8.9 | 112.5± 5.6 | 80.3±0.4 | 90.0±3.6 | 77.8±0.8 | 74.5±3.3 |
| ant-medium | 123.8±1.5 | 136.9±1.6 | 119.3±3.1 | 120.3±1.6 | 96.4±0.3 | 104.2±3.0 | 95.1±3.5 | 89.2±5.9 |
| *Total (Gym Mujoco)* | 316.8 | 354.8 | 284.9 | 297.2 | 222.3 | 241.1 | 215.5 | 205.6 |
| **Avg (Antmaze&Gym Mujoco)** | 497.7 | 537.1 | 458.0 | 478.7 | 391.9 | 430.6 | 398.5 | 371.1 |

online performance, necessitating comparative analysis. Specifically, we use Cal-QL and CQL as the selected offline-to-online algorithms and test them in combination with different exploration algorithms on the hardest task in the antmaze environment and some offline gym-mujoco tasks, while also documenting the results of online fine-tuning. As shown in Table 1, QCSE outperforms VCSE and RND on the selected tasks. We believe that the advantage of QCSE over VCSE on the selected tasks stems from the fact that QCSE can incorporate the state transitions of decisions into the intrinsic reward process, thereby avoiding excessive exploration of low-value state samples and further ensuring more stable training outcomes. Therefore, QCSE plays a crucial role in offline-to-online scenarios that require stable online finetuning.

**Can QCSE surpass previous efficient offline-to-online algorithms?** In order to more intuitively demonstrate the effectiveness of QCSE, we replaced QCSE with a series of past efficient offline-to-online algorithms and conducted comparisons. As shown in Figure 5, we select CQL as the base algorithm and aggregate it with QCSE, APL (Zheng et al., 2023), PEX (Zhang et al., 2023), SUNG (Guo et al., 2024) and BR (Lee et al., 2021b) to test on tasks of Antmaze and Gym-Mujoco (medium, medium-replay) domains, and CQL-QCSE archives the best performance (**85.6**) over all selected baselines, which demonstrating QCSE's competitive performance.

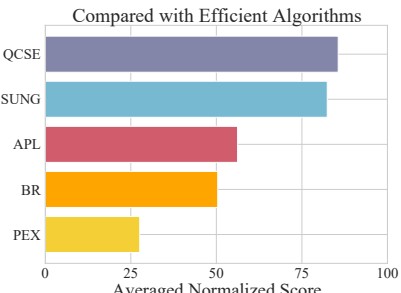

Figure 5: Performance Comparison. CQL+efficient offline-to-online approaches.

## 6.2 ABLATIONS

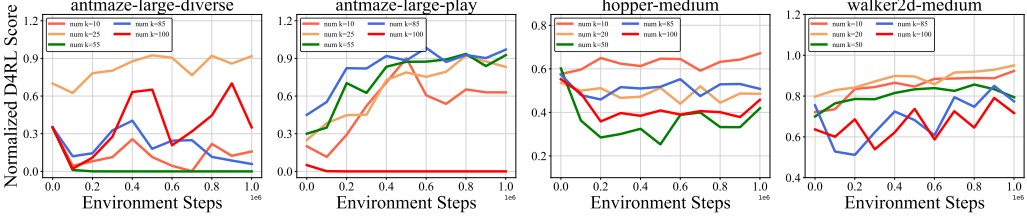

Figure 6: We evaluate the performance when varying the number of state clusters. We assess QCSE by configuring different sizes of k-nearest neighbor (knn) clusters and subsequently observe the impact of these parameter settings on online fine-tuning, and it can be observed that the choice of knn cluster settings exerts a notable influence on QCSE's performance.

**Effect of Hyperparameter.** We primarily investigate the effect of the hyperparameter in Equation 13. Specifically, the Q conditioned state entropy is approximated using the KSG estimator, where the number of state clusters serves as a crucial hyperparameter. As shown in Figure 6, the performance can indeed be influenced by the number of state clusters, and a trade-off exists among the sizes of these state clusters. For instance, the optimal cluster settings of walker2d and hopper are saturated around 20 and 10, respectively. In contrast, a task like amaze-large-diverse re-

quires a larger number of clusters (*about* 25). We consider the main reason is that different tasks require varying degrees of exploration, and thus need different cluster settings.

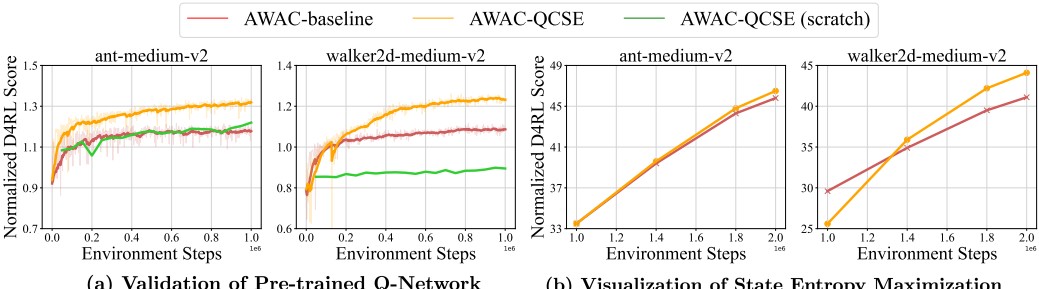

(a) **Validation of Pre-trained Q-Network**      (b) **Visualization of State Entropy Maximization**

Figure 7: (a) Ablation experiments to validate the impact of pre-trained Q network. (b) Quantitative results on the agent's state entropy.

**Scaled Ablations.** We utilize AWAC to further validate the effectiveness of QCSE. Specifically, we examine the impact of utilizing an offline pre-trained Q-network versus a randomly initialized Q-network to compute intrinsic rewards, as shown in Figure 7 (a). Offline pre-trained Q as condition performs better than randomly initialized Q as condition. On the other hand, in Figure 7 (b), we illustrate the progression of state entropy throughout the training process. Specifically, we can observe that the state entropy of AWAC-QCSE eventually exceeds AWAC baseline, indicating the influence of QCSE on state entropy.

**Solution for Acceleration.** Although our primary experiments have demonstrated the effectiveness of QCSE in offline to online tasks, using KNN to compute intrinsic rewards entails high computational complexity. Furthermore, computing intrinsic rewards at every step would result in significant consumption of computational resources. To reduce the computational resource consumption of QCSE, we can compute the intrinsic reward every n steps, thereby enhancing the training efficiency of QCSE. To validate the effectiveness of this approach, we selected CQL and CalQL for testing on the `pen-binary` task. Specifically, when n is set to 100, CalQL+QCSE can maintain the total training time close to the baseline while ensuring no obvious decrease in performance. When n is set to 20, CalQL+QCSE can maintain a performance that is better than CalQL, while only increasing the total training time by approximately one hour. When n is set to 50, it can keep the training time of CQL+QCSE at a relatively low level, while also ensuring that the policy performance does not degrade significantly (better than CQL). Therefore, calculating intrinsic rewards at fixed intervals of training steps can reduce training time without significantly compromising training performance.

Table 4: Training time by calculating the entropy rewards at regular intervals of training steps. We selected the pen-binary task for our testing and evaluated each task using a single Nvidia-RTX 2080 Ti GPU. Specifically, in this table, we label the algorithm that utilizes QCSE and calculates entropy at an interval of i steps as **n**(i), and we label *base* algorithms that donot utilize intrisic reward as base, **h** denotes hour.

| Algorithms | standard | *base* | **n**(1) | **n**(10) | **n**(20) | **n**(50) | **n**(100) |
|---|---|---|---|---|---|---|---|
| CalQL | score | 97 | 99 | 93 | 98 | 97 | 95 |
| | time | 6.296**h** | 21.65**h** | 8.082**h** | 7.393**h** | 6.856**h** | 6.682**h** |
| CQL | score | 20 | 99 | 89 | 0 | 89 | 0 |
| | time | 8.293**h** | 19.57**h** | 7.602**h** | 6.877**h** | 6.214**h** | 6.208**h** |

## 7 CONCLUSIONS

We propose a generalized offline-to-online framework called QCSE. **On the theoretical aspect**, we demonstrate that QCSE can implicitly realize SMM. Meanwhile, we showcase QCSE can guarantee the monotonicity of soft-Q optimization. **On the experimental aspect**, QCSE leads to improvements for both CQL and Cal-QL, validating our theoretical claims. We also extend the tests of QCSE to other model-free algorithms, and experimental results showed that QCSE performs better when combined with other model-free algorithms, demonstrating its generality.

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

CONTENTS

**Limitations and future work.**    Although we propose that maximizing state entropy can approximate State Marginal Matching (SMM) to ensure fine-tuned performance, we do not provide an explicit approach to address distribution shift issues (under offline-to-online setting). In the future, we aim to enhance QCSE to explicitly achieve SMM and assess whether this leads to improved fine-tuned performance.

## A   ETHICAL CLAIM

Despite the potential of offline RL to learn from the static datasets without the necessity to access the online environment, the offline method does not guarantee the optimal policy. Therefore, online fine-tuning is essential for policy improvement. In this study, we propose a novel and versatile reward augmentation framework, named Q conditioned state entropy maximization (QCSE) which can be seamlessly plugged into various model-free algorithms. We believe our approach is constructive and will enhance the sample efficiency of offline-to-online RL. Additionally, given that QCSE is an integrated algorithm, we also believe it can broadly and readily benefit existing algorithms.

## B   THEORETICAL ANALYSIS

In this section, we provide the supplementary mathematical analysis for QCSE.

### B.1   STATE ENTROPY MAXIMIZATION AND STATE MARGINAL MATCHING

In this section, we will discuss why maximizing state entropy contributes to achieving state marginal matching.

$$
\begin{aligned}
\max_{\substack{\rho_\pi(\mathbf{s}) \\ s.t. \max_\pi SoftQ^\pi}} \mathcal{H}[\mathbf{s}] &= \max_{\substack{\rho_\pi(\mathbf{s}) \\ s.t. \max_\pi SoftQ^\pi}} \int_{\mathbf{s}\sim\mathcal{S}} -\rho_\pi(\mathbf{s})\log\rho_\pi(\mathbf{s}) \\
&= \max_{\substack{\rho_\pi(\mathbf{s}) \\ s.t. \max_\pi SoftQ^\pi}} \int_{\mathbf{s}\sim\mathcal{S}_1} -\rho_\pi(\mathbf{s})\log\rho_\pi(\mathbf{s}) + \int_{\mathbf{s}\sim\mathcal{S}_2} -\rho_\pi(\mathbf{s})\log\rho_\pi(\mathbf{s}) \\
&\leq \max_{\substack{\rho_\pi(\mathbf{s}) \\ s.t. \max_\pi SoftQ^\pi}} \underbrace{\int_{\mathbf{s}\sim\mathcal{S}_2} -\rho_\pi(\mathbf{s})\log\rho_\pi(\mathbf{s})}_{\text{term2}} + \underbrace{\int_{\mathbf{s}\sim\mathcal{S}_1} -p^*(\mathbf{s})\log p^*(\mathbf{s})}_{\text{term1}},
\end{aligned}
\tag{4}
$$

where $p^*(\mathbf{s})$ denotes the target density, $\rho_{\pi(\mathbf{s})}$ denotes the marginal state distribution initialized by the offline dataset, as defined in Definition 1. $\mathcal{S}_1$ denotes domain where $\rho_\pi(\mathbf{s}) > p^*(\mathbf{s})|_{\mathbf{s}\sim\mathcal{S}_1}$, and $\mathcal{S}_2$ denotes domain where $\rho_\pi(\mathbf{s}) \leq p^*(\mathbf{s})|_{\mathbf{s}\sim\mathcal{S}_2}$. In this section, we examine the mathematical viability of the QCSE framework, focusing on two key aspects: **1)** Guarantee of Soft policy optimization **2)** Prevention of OOD state actions.

We first introduce the modified soft Q Bellman backup operator, denoted as Equation 5,

$$
\mathcal{T}^\pi_{QCSE}Q(\mathbf{s}_t, \mathbf{a}_t) \triangleq r(\mathbf{s}_t, \mathbf{a}_t) + r^{QCSE}(\mathbf{s}_t, \mathbf{a}_t) + \gamma\mathbb{E}_{\mathbf{s}_{t+1}\sim p}[V(\mathbf{s}_{t+1})]
\tag{5}
$$

In this equation, the term $V(\mathbf{s}_t) = \mathbb{E}_{\mathbf{a}_t\sim\pi}[Q(\mathbf{s}_t, \mathbf{a}_t) - \log\pi(\mathbf{a}_t\mid\mathbf{s}_t)]$ is defined.

**Lemma B.1** (Soft Policy Evaluation with QCSE.). *Given the modified soft bellman backup operator $\mathcal{T}^\pi_{QCSE}$ in Equation 5, along with a mapping $Q^0 : \mathcal{S} \times \mathcal{A} \to \mathbb{R}$ where $|\mathcal{A}| < \infty$. We define an iterative sequence as $Q^{k+1} = \mathcal{T}^\pi Q^k$. It can be shown that when index $k$ tends towards infinity, the sequence $Q^k$ converges to a soft Q-value of $\pi$.*

*proof.* Let us define the QCSE reward as follows

$$
r^\pi_{QCSE}(\mathbf{s}_t, \mathbf{a}_t) \triangleq r(\mathbf{s}_t, \mathbf{a}_t) + \lambda\operatorname{Tanh}\left(\mathcal{H}\left(\mathbf{s}_t\mid\min\left(Q_{\phi_1}(\mathbf{s}_t, \mathbf{a}_t), Q_{\phi_2}(\mathbf{s}_t, \mathbf{a}_t)\right)\right)\right) + \mathbb{E}_{\mathbf{s}_{t+1}\sim p}[\mathcal{H}(\pi(\cdot\mid\mathbf{s}_{t+1}))]
\tag{6}
$$

and rewrite the update rule as

$$Q\left(\mathbf{s}_t, \mathbf{a}_t\right) \leftarrow r^\pi_{QCSE}\left(\mathbf{s}_t, \mathbf{a}_t\right) + \gamma \mathbb{E}_{\mathbf{s}_{t+1} \sim p, \mathbf{a}_{t+1} \sim \pi}\left[Q\left(\mathbf{s}_{t+1}, \mathbf{a}_{t+1}\right)\right]. \tag{7}$$

Then we can apply mathematical analysis of convergence for policy evaluation as outlined in Sutton and Barto (1998) to prove the result. It is essential to note that the assumption $|\mathcal{A}| < \infty$ is necessary to ensure the boundedness of the QCSE reward."

**Lemma B.2** (Soft Policy Improvement with QCSE). *Let $\pi_{old} \in \Pi$, and let $\pi_{new}$ be the solution to the minimization problem defined as:*

$$\pi_{\text{new}} = \arg \min_{\pi' \in \Pi} D_{KL}\left(\pi'\left(\cdot \mid \mathbf{s}_t\right) \,\Big\|\, \frac{\exp\left(Q^{\pi_{\text{old}}}\left(\mathbf{s}_t, \cdot\right)\right)}{Z^{\pi_{\text{old}}}\left(\mathbf{s}_t\right)}\right). \tag{8}$$

*Then, it follows that $Q^{\pi_{new}}\left(\mathbf{s}_t, \mathbf{a}_t\right) \geq Q^{\pi_{old}}\left(\mathbf{s}_t, \mathbf{a}_t\right)$ for all $\left(\mathbf{s}_t, \mathbf{a}_t\right) \in \mathcal{S} \times \mathcal{A}$ provided that $|\mathcal{A}| < \infty$.*

*proof.* Starting from Equation 9, which has been established in the work by (Haarnoja et al., 2018), as:

$$\mathbb{E}_{\mathbf{a}_t \sim \pi_{\text{new}}}\left[Q^{\pi_{\text{old}}}\left(\mathbf{s}_t, \mathbf{a}_t\right) - \log \pi_{\text{new}}\left(\mathbf{a}_t \mid \mathbf{s}_t\right)\right] \geq V^{\pi_{\text{old}}}\left(\mathbf{s}_t\right), \tag{9}$$

we proceed to consider the soft Bellman equation, which can be expressed as:

$$\begin{aligned}
Q^{\pi_{\text{old}}}\left(\mathbf{s}_t, \mathbf{a}_t\right) &= r\left(\mathbf{s}_t, \mathbf{a}_t\right) + r_{QCSE}\left(\mathbf{s}_t, \mathbf{a}_t\right) + \gamma \mathbb{E}_{\mathbf{s}_{t+1} \sim p}\left[V^{\pi_{\text{old}}}\left(\mathbf{s}_{t+1}\right)\right] \\
&\leq r\left(\mathbf{s}_t, \mathbf{a}_t\right) + r_{QCSE}\left(\mathbf{s}_t, \mathbf{a}_t\right) + \gamma \mathbb{E}_{\mathbf{s}_{t+1} \sim p}\big[\mathbb{E}_{\mathbf{a}_{t+1} \sim \pi_{\text{new}}}\left[Q^{\pi_{\text{old}}}\left(\mathbf{s}_{t+1}, \mathbf{a}_{t+1}\right)\right. \\
&\quad\left. - \log \pi_{\text{new}}\left(\mathbf{a}_{t+1} \mid \mathbf{s}_{t+1}\right)\right] \\
&\;\;\vdots \\
&\leq Q^{\pi_{\text{new}}}\left(\mathbf{s}_t, \mathbf{a}_t\right)
\end{aligned} \tag{10}$$

Here, we have iteratively expanded $Q^{\pi_{\text{old}}}$ on the right-hand side by applying both the soft Bellman equation and the inequality from Equation 9.

**Theorem B.3** (Converged QCSE Soft Policy is Optimal). *Repetitive using Lemma 1 and Lemma 2 to any $\pi \in \Pi$ leads to convergence towards a policy $\pi^*$. And it can be proved that $Q^{\pi^*}\left(\mathbf{s}_t, \mathbf{a}_t\right) \geq Q^\pi\left(\mathbf{s}_t, \mathbf{a}_t\right)$ for all policies $\pi \in \Pi$ and all state-action pairs $\left(\mathbf{s}_t, \mathbf{a}_t\right) \in \mathcal{S} \times \mathcal{A}$, provided that $|\mathcal{A}| < \infty$.*

*proof.*

Let $\pi_i$ represent the policy at iteration $i$. According to Lemma 2, the sequence $Q^{\pi_i}$ exhibits a monotonic increase. Given that rewards and entropy and thus $Q$ are bounded from above for policies within the set $\Pi$, the sequence converges to a certain policy $\pi^*$. It is essential to demonstrate that $\pi^*$ is indeed an optimal policy. Utilizing a similar iterative argument as employed in the proof of Lemma 2, we can establish that $Q^{\pi^*}\left(\mathbf{s}_t, \mathbf{a}_t\right) > Q^\pi\left(\mathbf{s}_t, \mathbf{a}_t\right)$ holds for all $\left(\mathbf{s}_t, \mathbf{a}_t\right) \in \mathcal{S} \times \mathcal{A}$. In other words, the soft value associated with any other policy in $\Pi$ is lower than that of the converged policy. Consequently, $\pi^*$ is confirmed as the optimal policy within the set $\Pi$.

**Theorem B.4** (Conservative Soft Q values with QCSE). *By employing a double Q network, we ensure that in each iteration, the Q-value from the single Q network, denoted as $Q^{\pi_i}_{single\ Q}\left(\mathbf{s}_t, \mathbf{a}_t\right)$, is greater than or equal to the Q-value obtained from the double Q network, represented as $Q^{\pi_i}_{double\ Q}\left(\mathbf{s}_t, \mathbf{a}_t\right)$, for all $\left(\mathbf{s}_t, \mathbf{a}_t\right) \in \mathcal{S} \times \mathcal{A}$, where the action space is finite.*

*proof.* Let's begin by defining $\hat{Q}\left(\mathbf{s}_t, \mathbf{a}_t\right) = \min\left(Q_{\phi_1}\left(\mathbf{s}_t, \mathbf{a}_t\right), Q_{\phi_2}\left(\mathbf{s}_t, \mathbf{a}_t\right)\right)$. We then proceed to examine the difference between the augmented rewards in the context of QCSE for the single Q and

double Q networks:

$$r_{QCSE}(\mathbf{s}_t, \mathbf{a}_t | \hat{Q}(\mathbf{s}_t, \mathbf{a}_t)) - r_{QCSE}(\mathbf{s}_t, \mathbf{a}_t | Q(\mathbf{s}_t, \mathbf{a}_t))$$

$$= \sum_{i=0}^{N} \log 2 \max(||s_i - s_i^{knn}||, ||\hat{Q}(\mathbf{s}_t, \mathbf{a}_t) - \hat{Q}^{knn}(\mathbf{s}_t, \mathbf{a}_t)||) -$$

$$\sum_{i=0}^{N} \log 2 \max(||s_i - s_i^{knn}||, ||Q(\mathbf{s}_t, \mathbf{a}_t) - Q^{knn}(\mathbf{s}_t, \mathbf{a}_t)||)$$

$$= \log \frac{\prod_{i=0}^{N} \max(||s_i - s_i^{knn}||, ||\hat{Q}(\mathbf{s}_t, \mathbf{a}_t) - \hat{Q}^{knn}(\mathbf{s}_t, \mathbf{a}_t)||)}{\prod_{i=0}^{N} \max(||s_i - s_i^{knn}||, ||Q(\mathbf{s}_t, \mathbf{a}_t) - Q^{knn}(\mathbf{s}_t, \mathbf{a}_t))||} \tag{11}$$

$$\approx \log \frac{\prod_{i=0}^{N} \max(||s_i - s_i^{knn}||, \mathcal{H}(\hat{Q}))}{\prod_{i=0}^{N} \max(||s_i - s_i^{knn}||, \mathcal{H}(Q)||)}$$

$$\leq \log \frac{\prod_{i=0}^{N} \max(||s_i - s_i^{knn}||, \mathcal{H}(Q))}{\prod_{i=0}^{N} \max(||s_i - s_i^{knn}||, \mathcal{H}(Q))} = 0$$

Consequently, we establish that $r_{QCSE}(\mathbf{s}_t, \mathbf{a}_t | \hat{Q}(\mathbf{s}_t, \mathbf{a}_t)) \leq r_{QCSE}(\mathbf{s}_t, \mathbf{a}_t | Q(\mathbf{s}_t, \mathbf{a}_t))$. Now we consider the modified soft Bellman equation

$$Q_{\text{double Q}}^{\pi_i}(\mathbf{s}_t, \mathbf{a}_t)$$

$$= r(\mathbf{s}_t, \mathbf{a}_t) + r_{QCSE}(\mathbf{s}_t, \mathbf{a}_t | \hat{Q}(\mathbf{s}_t, \mathbf{a}_t)) + \gamma \cdot \mathbb{E}_{\mathbf{s}_{t+1} \sim p}[\hat{V}(\mathbf{s}_{t+1})]$$

$$= r(\mathbf{s}_t, \mathbf{a}_t) + r_{QCSE}(\mathbf{s}_t, \mathbf{a}_t | \hat{Q}(\mathbf{s}_t, \mathbf{a}_t)) + \gamma \cdot \mathbb{E}_{\mathbf{s}_{t+1} \sim p, \mathbf{a}_{t+1} \sim \pi} \left[ \hat{Q}(\mathbf{s}_{t+1}, \mathbf{a}_{t+1}) - \log \pi(\mathbf{a}_{t+1} \mid \mathbf{s}_{t+1}) \right]$$

$$\vdots$$

$$= r(\mathbf{s}_t, \mathbf{a}_t) + r_{QCSE}(\mathbf{s}_t, \mathbf{a}_t | \hat{Q}(\mathbf{s}_t, \mathbf{a}_t)) + \gamma \cdot \mathbb{E}_{\mathbf{s}_{t+1} \sim p, \mathbf{a}_{t+1} \sim \pi}[r^{mod}(\mathbf{s}_{t+1}, \mathbf{a}_{t+1} | \hat{Q}(\mathbf{s}_{t+1}, \mathbf{a}_{t+1}))] \cdots +$$

$$\gamma^n \cdot \mathbb{E}_{\mathbf{s}_{t+n} \sim p, \mathbf{a}_{t+n} \sim \pi}[r^{mod}(\mathbf{s}_{t+n}, \mathbf{a}_{t+n} | \hat{Q}(\mathbf{s}_{t+n}, \mathbf{a}_{t+n}))] + \cdots + \text{entropy terms}$$

$$\leq r(\mathbf{s}_t, \mathbf{a}_t) + r_{QCSE}(\mathbf{s}_t, \mathbf{a}_t | Q(\mathbf{s}_t, \mathbf{a}_t)) + \gamma \cdot \mathbb{E}_{\mathbf{s}_{t+1} \sim p, \mathbf{a}_{t+1} \sim \pi}[r^{mod}(\mathbf{s}_{t+1}, \mathbf{a}_{t+1} | Q(\mathbf{s}_{t+1}, \mathbf{a}_{t+1}))] \cdots +$$

$$\gamma^n \cdot \mathbb{E}_{\mathbf{s}_{t+n} \sim p, \mathbf{a}_{t+n} \sim \pi}[r^{mod}(\mathbf{s}_{t+n}, \mathbf{a}_{t+n} | Q(\mathbf{s}_{t+n}, \mathbf{a}_{t+n}))] + \cdots + \text{entropy terms}$$

$$= Q_{\text{single Q}}^{\pi_i}(\mathbf{s}_t, \mathbf{a}_t)$$

$$\tag{12}$$

where we have repeatedly expanded $\hat{Q}$ in terms of QCSE rewards to obtain the final inequality $Q_{\text{singleQ}}^{\pi_i} \geq Q_{\text{double Q}}^{\pi_i}$.

## C  EXPERIMENTAL SETUP

In this section, we introduce the benchmarks and dataset we utilized, specifically, we mainly utilize Gym-Mujoco and antmaze to test our algorithm.

### C.1  GYM-MUJOCO

Our benchmars from Gym-Mujoco domain mainly includes `halfcheetah`, `ant`, `hopper` and `walker2d`, and concrete information of these benchmarks can be referred to table 5. In paticular, the action and observation space of these locomotion benchmarks are continuous and any decision making will receive an immediate reward.

### C.2  ANTMAZE

Our benchmars from antmaze mainly includes `antmaze-large-diverse`, `antmaze-medium-diverse`, `antmaze-large-play` and `antmaze-medium-play`, concrete information of our benchmarks can be referred to table 6.

| Environment | Task Name | Samples | Observation Dim | Action Dim |
|---|---|---|---|---|
| halfcheetah | medium | $10^6$ | 6 | 17 |
| walker2d | medium | $10^6$ | 6 | 17 |
| hopper | medium | $10^6$ | 3 | 11 |
| ant | medium | $10^6$ | 8 | 111 |
| halfcheetah | medium-replay | $2.02 \times 10^5$ | 6 | 17 |
| walker2d | medium-replay | $3.02 \times 10^5$ | 6 | 17 |
| hopper | medium-replay | $4.02 \times 10^5$ | 3 | 11 |
| ant | medium-replay | $3.02 \times 10^5$ | 8 | 111 |

Table 5: Introduction of D4RL tasks (Gym-Mujoco).

| Environment | Task Name | Samples | Observation Dim | Action Dim |
|---|---|---|---|---|
| antmaze | large-diverse | $10^6$ | 29 | 8 |
| antmaze | large-play | $10^6$ | 29 | 8 |
| antmaze | medium-diverse | $10^6$ | 29 | 8 |
| antmaze | medium-play | $10^6$ | 29 | 8 |

Table 6: Introduction of D4RL tasks (Antmaze).

## D  IMPLANTATION DETAILS

### D.1  OFFLINE-TO-ONLINE IMPLANTATION

The workflow of our method is similar to the most of offline-to-online algorithms that we firstly pre-train on offline datasets, followed by online fine-tuning (Interacting with online environment to collect online dataset and followed by fine-tuning on offline and online datasets).

### D.2  EVALUATION DETAILS

Our evaluation method can be refered to Fu et al. (2021). That is for each evaluation, we freeze the parameter of trained model, and then conducting evaluation $10 \sim 50$ times and then computing the normalized score via $\frac{\text{score}_{\text{evaluation}} - \text{score}_{\text{expert}}}{\text{score}_{\text{expert}} - \text{score}_{\text{random}}}$, and then averaging these normalized evaluation scores.

### D.3  QCSE IMPLANTATION

In QCSE framework, we modify our reward as :

$$r^{\text{mod}}(\mathbf{s}, \mathbf{a}) = \lambda \cdot \underbrace{\text{Tanh}(\mathcal{H}(\mathbf{s}|\min(Q_{\phi_1}(\mathbf{s}, \mathbf{a}), Q_{\phi_2}(\mathbf{s}, \mathbf{a}))))}_{r^{\text{mod}}} + r(\mathbf{s}, \mathbf{a}), \quad (\mathbf{s}, \mathbf{a}) \sim \mathcal{D}_{\text{online}} \quad (13)$$

To calculate the intrinsic reward $r^{\text{mod}}$ for the online replay buffer $D_{\text{online}}$, we use the KSG estimator, as defined in Equation 14, to estimate the conditional state density of the empirical dataset $D_{\text{online}}$

$$r^{\text{mod}}(\mathbf{s}, \mathbf{a}) = \frac{1}{d_s}\phi(n_v(i)+1) + \log 2 \cdot \max(||\mathbf{s}_i - \mathbf{s}_i^{knn}||, ||\hat{Q}(\mathbf{s}, \mathbf{a}) - \hat{Q}(\mathbf{s}, \mathbf{a})^{knn}||), (\mathbf{s}, \mathbf{a}) \sim \mathcal{D}_{\text{online}}. \quad (14)$$

Given that the majority of our selected baselines are implemented using the double Q($\{Q_{\phi_1}, Q_{\phi_2}\}$), the offline pre-trained double Q can be readily utilized for the computation of intrinsic rewards, and we found that the performance of QCSE is sutured when $\lambda$ is set to 1. We also provide a (Variance Auto Encoder) VAE implantation (Equation 15) of QCSE, this realization is computing efficiency, but require extraly training a VAE model, due to Equation 3 won't require training thus we mainly test Equation 3.

$$r^{\text{mod}}(\mathbf{s}, \mathbf{a}) = -\log p_{\hat{\phi}}(s|\hat{Q}(\mathbf{s}, \mathbf{a})) = -\log \mathbb{E}_{z \sim q_\phi(z|\mathbf{s}, \hat{Q}(\mathbf{s}, \mathbf{a}))}[\frac{p_{\hat{\phi}}(\mathbf{s}|\hat{Q}(\mathbf{s}, \mathbf{a}))}{q_\phi(z|\mathbf{s}, \hat{Q}(\mathbf{s}, \mathbf{a}))}], (\mathbf{s}, \mathbf{a}) \sim \mathcal{D}_{\text{online}}. \quad (15)$$

We will test and compare the performance difference and computing efficiency between Equation 15 and Equation 14 in the future.

### D.4 CODEBASE

Our implementation is based on Cal-QL:https://github.com/nakamotoo/Cal-QL, VCSE:https://sites.google.com/view/rl-vcse. Additionally, we have included our source code in the supplementary material for reference. Readers can refer to our pseudocode (see Algorithm 1) for a comprehensive understanding of the implementation details. $\hat{Q}$ see [2]. Part of our source code will be released at: .

---

**Algorithm 1** Training QCSE

---

**Require:** Pre-collected data $\mathcal{D}_{\text{offline}}$.
 1: Initialize $\pi_\theta$, and $Q_{\phi_1}, Q_{\phi_2}$.
    // Offline Pre-training Stage.
 2: **for** $k = 1, \cdots, K$ **do**
 3:    Learn $Q_\phi$ on $\mathcal{D}_{\text{offline}}$ by Equation 17 or 16 //We compute target Q value via $Q_{\text{target}}$, learning $Q_{\text{target}}$ by
       Empirical Momentum Average (EMA),*i.e.* $Q_{\text{target}} = (1 - \alpha)Q_\phi + \alpha Q_{\text{target}}$.
 4:    Learn $\pi_\theta$ on $\mathcal{D}_{\text{offline}}$ with Equation 18.
 5: **end for**
    // Online Fine-tuning Stage.
 6: **for** $k = 1, \cdots, K$ **do**
 7:    Interacting $\pi_\theta$ to obtain $\mathcal{D}_{\text{online}}$.
 8:    Augmenting Reward in $\mathcal{D}_{\text{online}}$ by Equation 2.
 9:    Sample a batch offline data $\mathcal{D}_{\text{offline}}$, and build training batch,*i.e.* $\mathcal{D}_{\text{mix}} = \mathcal{D}_{\text{offline}} \cup \mathcal{D}_{\text{online}}$ //mixture of
       offline and online is not necessary required, it depends on the quality of offline dataset.
10:    Learn $\pi_\theta, Q_{\phi_1}$, and $Q_{\phi_2}$ on $\mathcal{D}_{\text{mix}}$ with the same objective in offline stage.
11: **end for**

---

$$\mathcal{L}(Q) = \mathbb{E}_{(\mathbf{s},\mathbf{a})\sim\mathcal{D}}[(Q(\mathbf{s},\mathbf{a})-\mathcal{B}_\mathcal{M}^\pi Q(\mathbf{s},\mathbf{a}))^2]+\mathrm{E}_{\mathbf{s}\sim\mathcal{D},\mathbf{a}\sim\pi}[\max(Q(\mathbf{s},\mathbf{a}),V^\mu(\mathbf{s}))]-\mathrm{E}_{(\mathbf{s},\mathbf{a})\sim\mathcal{D}}[Q(\mathbf{s},\mathbf{a})]. \tag{16}$$

$$\mathcal{L}(Q) = \mathbb{E}_{(\mathbf{s},\mathbf{a},\mathbf{s}')\sim\mathcal{D}}[(Q(\mathbf{s},\mathbf{a}) - \mathcal{B}_\mathcal{M}^\pi Q(\mathbf{s},\mathbf{a}))^2] + \mathbb{E}_{(\mathbf{s},\mathbf{a},\mathbf{s}')\sim\mathcal{D}}[-Q(\mathbf{s},\mathbf{a}) + Q(\mathbf{s}',\pi(\mathbf{s}'))], \tag{17}$$

$$\mathcal{J}(\pi_\theta) = \mathrm{E}_{\mathbf{s}\sim\mathcal{D}}[-Q(\mathbf{s},\pi_\theta(\mathbf{s})) + \alpha \log(\pi_\theta(\mathbf{s}))]. \tag{18}$$

### D.5 COMPUTING RESOURCES

Our experiments were run on a computer cluster with 4×32GB RAM, AMD EPYC 7742 64-Core CPU, and NVIDIA-A100 GPU, Linux. Most of our code base (The implantation of **Cal-QL**, **CQL**, **TD3+BC**, **SAC**) are based on JAX [3], part of our implantation (**IQL**, **AWAC**) are based on Pytorch[4] (We use different deep learning frameworks mainly to preliminary validate that our algorithm can work in various of deep learning frameworks).

### D.6 OUR HYPER-PARAMETER

**Hyper-parameter of QCSE.** The K-nearest neighbors (knn) for QCSE are configured as follows: [0, 10, 15, 25, 50, 85, 100, 110], and the parameter $\lambda$ in Equation 13 is set to 1.

**Hyper-parameter of Baselines** In the context of these algorithms, we conducted tests related to AWAC and IQL using the repository available at https://github.com/tinkoff-ai/CORL, while tests related to Cal-QL and CQL were performed using the repository accessible at https://github.com/nakamotoo/Cal-QL. The following five tables present fundamental but critical hyperparameter settings for five baseline algorithms.

---

[2]where $\phi_1$ and $\phi_2$ are the params of double Q Networks and $\hat{Q}(\mathbf{s},\mathbf{a}) = \min(Q_{\phi_1}(\mathbf{s},\mathbf{a}), Q_{\phi_2}(\mathbf{s},\mathbf{a}))$, and $x_i^{knn}$ is the $n_x(i)$-th nearest neighbor of $x_i$.

[3]https://github.com/google/jax.git
[4]https://pytorch.org/

Table 7: Hyper-parameters of AWAC.

| Hyperparameter | Value |
|---|---|
| 0ffline pre-train iterations | $1e^6$ |
| 0nline fine-tuning iterations | $1e^6$ |
| Buffer size | 20000000 |
| Batch size | 256 |
| learning rate | $3e^{-4}$ |
| $\gamma$ | 0.99 |
| awac $\tau$ | 5e-3 |
| awac $\lambda$ | 1.0 |
| Actor Architecture | $4\times$ Layers MLP (hidden dim 256) |
| Critic Architecture | $4\times$ Layers MLP (hidden dim 256) |

Table 8: Hyper-parameters of IQL.

| Hyperparameter | Value |
|---|---|
| 0ffline pre-train iterations | $1e^6$ |
| 0nline fine-tuning iterations | $1e^6$ |
| Batch size | 256 |
| learning rate of $\pi$ | $3e^{-4}$ |
| learning rate of V | $3e^{-4}$ |
| learning rate of Q | $3e^{-4}$ |
| $\gamma$ | 0.99 |
| IQL $\tau$ | 0.7 # *Coefficient for asymmetric loss* |
| $\beta$ (Inverse Temperature) | 3.0# *small beta $\rightarrow$ BC, big beta $\rightarrow$ maximizing Q* |
| Actor Architecture | $4\times$ Layers MLP (hidden dim 256) |
| Critic Architecture | $4\times$ Layers MLP (hidden dim 256) |

Table 9: Hyper-parameters of TD3+BC.

| Hyperparameter | Value |
|---|---|
| 0ffline pre-train iterations | $1e^6$ |
| 0nline fine-tuning iterations | $1e^6$ |
| learning rate of $\pi$ | $1e^{-4}$ |
| learning rate of Q | $3e^{-4}$ |
| $\gamma$ | 0.99 |
| Batch size | 256 |
| TD3 alpha | 2.5 |
| Actor Architecture | $4\times$ Layers MLP (hidden dim 256) |
| Critic Architecture | $4\times$ Layers MLP (hidden dim 256) |

Table 10: Hyper-parameters of Cal-QL. We only provide the basic setting, for more detail setting, please directly refer to https://nakamotoo.github.io/projects/Cal-QL

| Hyperparameter | Value |
|---|---|
| 0ffline pre-train iterations | $1e^6$ |
| 0nline fine-tuning iterations | $1e^6$ |
| learning rate of $\pi$ | $1e^{-4}$ |
| learning rate of Q | $3e^{-4}$ |
| $\gamma$ | 0.99 |
| Batch size | 256 |
| Actor Architecture | $4\times$ Layers MLP (hidden dim 256) |
| Critic Architecture | $4\times$ Layers MLP (hidden dim 256) |

Table 11: Hyper-parameters of CQL. CQL uses Cal-QL's code-base, and we only need to remove Cal-QL's calibration loss when deploying CQL.

| Hyperparameter | Value |
|---|---|
| Offline pre-train iterations | $1e^6$ |
| Online fine-tuning iterations | $1e^6$ |
| learning rate of $\pi$ | $1e^{-4}$ |
| learning rate of Q | $3e^{-4}$ |
| $\gamma$ | 0.99 |
| Batch size | 256 |
| Actor Architecture | $4\times$ Layers MLP (hidden dim 256) |
| Critic Architecture | $4\times$ Layers MLP (hidden dim 256) |

# E  APPENDED EXPERIMENTAL RESULTS

In Table 12, we compare a series of different efficient offline-to-online methods, including APL, PEX, and BR. Specifically, we tested these methods on the ant-maze domain and the `medium` and `medium-replay` tasks in the Gym-Mujoco environment. We found that QCSE shows the best overall performance, indicating that QCSE, when paired with CQL, can achieve superior results.

Table 12: Comparison of various efficient offline-to-online methods. Part of experimental results are quoted from Guo et al. (2024).

| Task | CQL+APL | CQL+PEX | CQL+BR | CQL+SUNG | CQL+QCSE |
|---|---|---|---|---|---|
| antmaze-large-diverse | 0 | 0 | 0.1 | 44.1 | 89.8 |
| antmaze-large-play | 0 | 0 | 0 | 52.7 | 92.6 |
| antmaze-medium-diverse | 36.8 | 0.3 | 13.6 | 85.6 | 98.9 |
| antmaze-medium-play | 22.8 | 0.3 | 22.2 | 86.3 | 99.4 |
| halfcheetah-medium | 44.7 | 43.5 | 56.7 | 79.7 | 87.9 |
| walker2d-meidum | 75.3 | 34.0 | 81.7 | 86.0 | 130.0 |
| hopper-medium | 102.7 | 46.3 | 97.7 | 104.1 | 62.4 |
| halfcheetah-medium-replay | 78.6 | 45.5 | 64.9 | 75.6 | 55.4 |
| walker2d-medium-replay | 103.2 | 40.1 | 88.5 | 108.2 | 112.7 |
| hopper-medium-replay | 97.4 | 66.5 | 78.8 | 101.9 | 27.1 |
| **Average Fine-tuned** | 56.2 | 27.6 | 50.4 | 82.4 | 85.6 |

## F EXTENDED EXPERIMENTS

### F.1 THE TRAINING PERFORMANCE OF QCSE ON MEDIUM-EXPERT

To further analyze the impact of the offline dataset on the experimental performance of QCSE, we compared the performance of CQL and CQL+QCSE on the medium-expert dataset, respectively. We found that QCSE did not significantly improve CQL on the medium-expert dataset, similar to its performance on the medium dataset. This demonstrates that our algorithm has already converged on the medium dataset and can effectively enhance the fine-tuning effect of the pre-training strategy on suboptimal datasets.

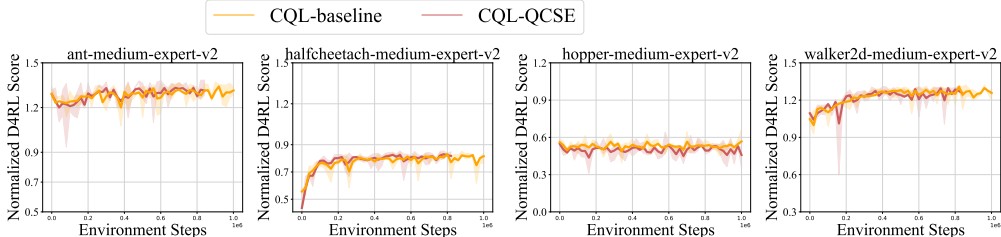

Figure 8: We compare the performance of CQL+QCSE and CQL on the medium-expert dataset.

**State Entropy as Intrinsic Reward.** If the state density $\rho(\mathbf{s})$ is unknown, we can instead using non-parametric entropy estimator to approximate the state entropy (Seo et al., 2021). Specifically, given N i.i.d. samples $\{\mathbf{s}_i\}$, the k-nearest neighbors (knn) entropy estimator can be defined as[5]:

$$\hat{H}_N^k(S) = \frac{1}{N}\sum_{i=1}^{N} \log \frac{N \cdot ||\mathbf{s}_i - s_i^{knn}||_2^{d_\mathbf{s}} \cdot \hat{n}_{\hat{\pi}}^{\frac{d_\mathbf{s}}{2}}}{k \cdot \Gamma(\frac{d_\mathbf{s}}{2} + 1)} \propto \frac{1}{N}\sum_{i=1}^{N} \log ||\mathbf{s}_i - \mathbf{s}_i^{knn}||. \tag{19}$$

**Visualization of State Entropy Changing.** In this experiment, for each training step, we select the buffer and randomly sample 5000 instances to approximate the entropy using Equation 10. and then plot the trend of approximated state entropy. For the majority of the tasks, the state entropy of *AWAC-QCSE* was either progressively greater than or consistently exceeded that of *AWAC-base*. This indicates that QCSE effectively enhances the agent's exploratory tendencies, enabling them cover much more observation region.

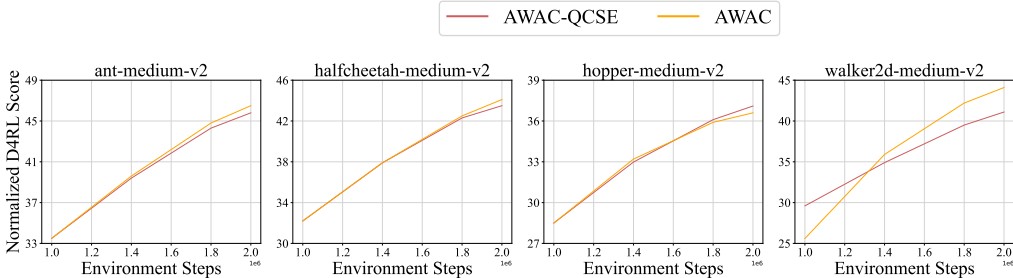

Figure 9: The Changing of Approximated Entropy along with increasing training steps. We found that the approximated state entropy in the buffer collected by AWAC using QCSE was greater in the later stages of online finetuning.

### F.2 PRE-TRAINED Q VS. RANDOM Q

**Pre-trained Q condition versus un-pre trained Q condition.** To validate the statement in our main paper that intrinsic reward computation is influenced by the initialization of $Q$, we conducted

---

[5]$d_s$ is the dimension of state and $\Gamma$ is the gamma function, $\hat{n}_{\hat{\pi}} \propto 3.14$.

experiments comparing the effects of pre-trained initialized $Q$ and from-scratch[6] trained $Q$ during intrinsic reward calculation. Our findings indicate that intrinsic rewards based on offline-initialized $Q$ generally outperform those derived from a from-scratch trained $Q$ across most tasks.

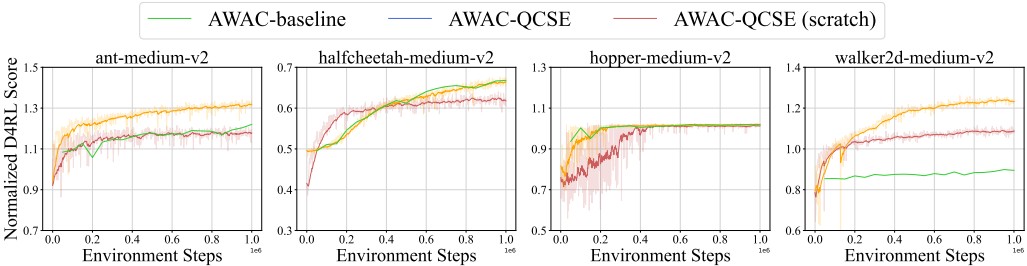

Figure 10: Offline Pre-trained Q condition vs. Randomly initialized Q condition. In the majority of our selected Gym-Mujoco tasks, the use of offline-initialized intrinsic reward conditions yielded better performance and higher sample efficiency. To provide clarity, *AWAC-base* means AWAC algorithm without any modification, *AWAC-QCSE* signifies AWAC with QCSE augmentation, and *AWAC-QCSE (scratch)* denotes AWAC with QCSE where the computation of reward conditions satisfying note 6

## F.3 AGGREGATED METRIC WITH QCSE

To demonstrate the significant improve brought by QCSE we adapt the method proposed by Agarwal et al., QCSE can significantly improve the performance of CQL and Cal-QL.

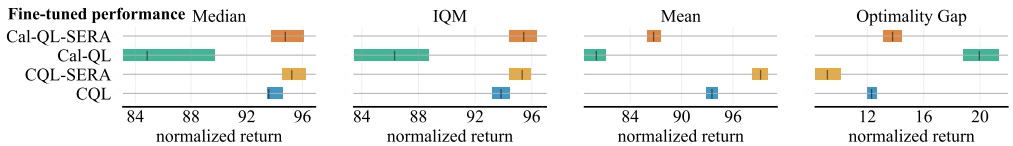

Figure 11: Aggregate metrics with QCSE (SERA). We refer to Agarwal et al. (2022) to analyze QCSE's performance. Specifically, higher median, IQM, and mean scores are better, QCSE can significantly improve the performance of CQL and Cal-QL.

## G EXTENDED RELATED WORK

### G.1 OFFLINE-TO-ONLINE RL

In this section, we systematically introduce recent developments in offline-to-online learning and summarize the corresponding methods, the first perspective involves adopting a conservative policy optimization during online fine-tuning, typically achieved through the incorporation of policy constraints. Specifically, there are three main approaches within this category. The first approach constrains the predictions of the fine-tuning policy within the scope of offline support during online fine-tuning (Wu et al., 2022). While this method contributes to achieving stable online fine-tuning performance, it tends to lead to overly conservative policy learning, and the accuracy of the estimation of offline support also influences the effectiveness of online fine-tuning. The second approach utilizes an offline dataset to constrain policy learning (Nair et al., 2021; Kostrikov et al., 2021; Xiao et al., 2023; Mark et al., 2023). However, the effectiveness of fine-tuning cannot be guaranteed if the dataset quality is poor. This method is sensitive to the quality of the dataset. The third approach employs pre-trained policies to constrain online fine-tuning, but this paradigm is influenced by the quality of the pre-trained policy (Zhang et al., 2023; Yu and Zhang, 2023). The second perspective involves adopting a conservative approach during offline training, specifically using pessimistic constraints to learn Q to avoid OOD (Out-of-Distribution) issues. Research in this category primarily

---

[6]We use from-scratch $Q$ to compute intrinsic reward, while continuing to utilize the offline-initialized $Q$ for conducting online fine-tuning.

includes: Learning a conservative Q during offline pre-training and employing an appropriate experience replay method during online learning or using Q ensemble during offline pre-training to avoid OOD issues (Lee et al., 2021a; Lyu et al., 2022; Hong et al., 2023). However, as this approach introduces conservative constraints during critic updates, the value estimates between offline and online are not aligned, leading to a decrease in performance during early online fine-tuning. Therefore, Cal-QL introduces a calibrated conservative term to ensure standard online fine-tuning (Nakamoto et al., 2023). Addtionally, there are also some other methods, such that ODT (Zheng et al., 2022) combined sequence modeling with Goal conditioned RL to conduct offline-to-online RL.

