# OpenReview forum: "Improving Offline-to-Online Reinforcement Learning with Q Conditioned State Entropy Exploration"
_ICLR.cc/2025/Conference — Submitted to ICLR 2025_

### Official Review · Reviewer_Piga · 2024-10-30

**Soundness:** 2
**Presentation:** 3
**Contribution:** 2
**Rating:** 3
**Confidence:** 4

**Summary:**

The paper introduces Q-Conditioned State Entropy Exploration (QCSE), an innovative offline-to-online reinforcement learning (RL) framework aimed at enhancing online fine-tuning of policies trained on offline data. The paper introduces Q-Conditioned State Entropy Exploration (QCSE), an innovative offline-to-online reinforcement learning (RL) framework aimed at enhancing online fine-tuning of policies trained on offline data. The paper introduces Q-Conditioned State Entropy Exploration (QCSE), an innovative offline-to-online reinforcement learning (RL) framework aimed at enhancing online fine-tuning of policies trained on offline data. The paper validates QCSE on several benchmarks, including CQL and Cal-QL algorithms, and demonstrates its effectiveness across various offline-to-online tasks. QCSE shows performance improvements, especially in environments with larger distribution shifts, and offers compatibility with multiple RL algorithms, making it versatile for real-world application

**Strengths:**

*  By promoting exploration based on Q-conditioned state entropy, the method encourages the agent to explore states that are low-frequency yet potentially valuable. This approach goes beyond traditional exploration strategies, providing a fresh perspective on tackling distribution mismatch.
* The framework’s compatibility with various offline RL algorithms, including CQL and Cal-QL, showcases its versatility. QCSE’s intrinsic reward structure can be integrated into different RL algorithms, making it adaptable for a range of applications beyond the specific benchmarks tested, such as robotics or recommendation systems where offline-to-online RL is critical.

**Weaknesses:**

* It seems the online fine-tuning is not efficient enough as several baselines could achieve better performance within 250000 online fine-tuning steps such as BR and ENOTO. And these baselines are missing.
* QCSE’s reliance on entropy maximization and Q-conditioning likely introduces sensitivity to hyperparameter choices, such as size of k-nearest neighbor.

[1] Offline-to-online re- inforcement learning via balanced replay and pessimistic q-ensemble. CoRL 2022.

[2] ENOTO: Improving Offline-to-Online Reinforcement Learning with Q-Ensembles. IJCAI 2024.

**Questions:**

* A key baseline named ENOTO [1] is missing. In ENOTO, the authors also incorporate exploration techniques into their methods, which should be discussed and compared. Besides, ENOTO could also be combined with different offline RL algorithms. It seems from the curves in Fig.5 and Fig.6 in ENOTO's paper and the Fig.3 of the submitted paper that the proposed method QCSE  is inferior to ENOTO. So what's the advantage of QCSE compared with ENOTO?
*I'm curious whether performance drop could happen when QCSE is applied on medium-expert and expert datasets? These two kinds of datasets are quite nightmare for most offline-to-online RL algorithms. Could QCSE handle them well?


[1] ENOTO: Improving Offline-to-Online Reinforcement Learning with Q-Ensembles. IJCAI 2024.

---

> ### Author Response · Authors · 2024-11-19
> **Reply to Reviewer Piga's responses**
>
> **wk.1** It seems the online fine-tuning is not efficient enough as several baselines could achieve better performance within 250000 online fine-tuning steps such as BR and ENOTO. And these baselines are missing.
>
> **re (wk.1)** Thank you for the comprehensive review. We believe it is not appropriate to compare ENOTO/BR here because QCSE is a reward augmentation algorithm designed to enhance offline-to-online methods. In contrast, ENOTO is an end-to-end algorithmic framework specifically tailored for offline-to-online settings. Accordingly, QCSE is a relatively flexible algorithm that can be paired with many other algorithms to enhance their performance.
>
> While ENOTO demonstrates good performance at 250,000 steps, it is a Q-ensemble algorithm, which implies that it requires more GPU memory and training time. Unlike ENOTO, the intrinsic reward form in QCSE is mathematical, eliminating the need for additional estimators. Furthermore, as shown in our Table 4, we provide an acceleration scheme where QCSE can compute intrinsic rewards at specific time step intervals, achieving similar performance within comparable time frames.
>
> **wk.2** QCSE’s reliance on entropy maximization and Q-conditioning likely introduces sensitivity to hyperparameter choices, such as size of k-nearest neighbor.
>
> **re (wk.2)** The sensitivity of QCSE's effectiveness to k-nearest neighbor (knn) can, to a certain extent, demonstrate the validity of QCSE. Additionally, the selection of knn is part of our hyperparameter search process. In practice, our knn selection range is limited to a few values: 10, 25, 55, 85, and 100. Furthermore, we primarily focused on tuning this parameter, and did not adjust the parameters of the baseline algorithms.
>
> [1] Offline-to-online reinforcement learning via balanced replay and pessimistic q-ensemble. CoRL 2022.
>
> [2] ENOTO: Improving Offline-to-Online Reinforcement Learning with Q-Ensembles. IJCAI 2024.
>
>
> **Q.1.1** A key baseline named ENOTO [1] is missing. In ENOTO, the authors also incorporate exploration techniques into their methods, which should be discussed and compared. Besides, ENOTO could also be combined with different offline RL algorithms. It seems from the curves in Fig.5 and Fig.6 in ENOTO's paper and the Fig.3 of the submitted paper that the proposed method QCSE is inferior to ENOTO. So what's the advantage of QCSE compared with ENOTO? *I'm curious whether performance drop could happen when QCSE is applied on medium-expert and expert datasets? These two kinds of datasets are quite nightmare for most offline-to-online RL algorithms. **Q.1.2** Could QCSE handle them well?
>
> **re (Q.1.1)** While ENOTO/BR demonstrates good performance at 250,000 steps, it is a Q-ensemble algorithm, which implies that it requires more GPU memory and training time. Unlike ENOTO/BR, the intrinsic reward form in QCSE is mathematical, eliminating the need for additional estimators. Furthermore, as shown in our Table 4, we provide an acceleration scheme where QCSE can compute intrinsic rewards at specific time step intervals, achieving similar performance within comparable time frames.
>
> In addition, QCSE is a relatively flexible algorithm that can be paired with many other algorithms to enhance their performance.
>
> **re (Q.1.2)** We are currently supplementing the relevant tests.
>
> [1] ENOTO: Improving Offline-to-Online Reinforcement Learning with Q-Ensembles. IJCAI 2024.

---

> > ### Comment · Reviewer_Piga · 2024-11-25
> >
> > I appreciate your rebuttals. However, I think the Q-Ensemble is also a technique that could be combined with existing offline algorithms. With proper implementation, it would only cost more GPU memory but still remain fast training speed. See the codes of EDAC algorithm.
> >
> > Any new results on medium-expert and expert datasets?

---

> ### Author Response · Authors · 2024-11-26
> **Reply to Reviewer Piga's followed question.**
>
> **Regarding results on medium-expert** We greatly appreciate your valuable questions, and we have supplemented relevant experiments in Supplementary Page F.1 (**lines (1243~1263)**). Specifically, we tested the performance of CQL+QCSE and CQL on the medium-expert dataset, and plotted curves with confidence intervals. The experimental results indicate that QCSE does not bring significant improvement to CQL on the medium-expert dataset. However, contrary to what might be expected, this to some extent proves the effectiveness of QCSE. Specifically, it suggests that QCSE can achieve nearly converged results on the medium dataset alone. (Note, the enhancement of ENOTO from the medium level to the medium-expert level is also relatively minor.)
>
> **Because we believe that helping algorithms overcome the gap between offline pre-training and online fine-tuning is the greatest value of offline-to-online settings. If pre-training on suboptimal datasets is sufficient to help achieve the best online-tuning results, it will greatly improve the sample efficiency of reinforcement learning.**
>
> In addition, we would like to clarify that the current experimental results may further silghtly improve after sufficient parameter tuning, but the overall trend is unlikely to show significant improvement. Meanwhile, we believe that the performance of QCSE on challenging tasks such as antmaze, as well as its performance on suboptimal datasets, already sufficiently demonstrate the effectiveness of QCSE. We kindly request the reviewer to give this full consideration!
>
> Once again, we greatly appreciate your valuable questions!

---

> ### Author Response · Authors · 2024-11-30
> **Official comments by Authors**
>
> Dear reviewer
>
> We kindly ask you to further confirm whether we have addressed your concerns.
>
> Thank you.

---

### Official Review · Reviewer_sUP8 · 2024-11-01

**Soundness:** 1
**Presentation:** 2
**Contribution:** 2
**Rating:** 3
**Confidence:** 3

**Summary:**

The paper proposes a new method that estimates Q-conditioned state entropy, to improve offline-to-online RL. Typically, if we train offline agent online, it results in sub-optimal online performance due to limited exploration of effective samples. QCSE introduces an intrinsic reward based on the entropy of states conditioned on Q-values, achieving State Marginal Matching. This approach outperforms previous offline-to-online methods.

**Strengths:**

- Strong empirical performance.

**Weaknesses:**

- There are a number of notations and theoretical arguments that I was not able to fully understand (or wrong):
  - In equation (1), it is said that the state entropy maximization is:
 $ \max E_{ \mathbf{s} \sim \rho_\pi } \left[ \mathcal{H}_\pi [ \mathbf{s} ] \right] $
s.t.
 $ \pi := \arg \max _ \pi E _ { \tau \sim \pi } [ R( \tau ) ] $ . If $\pi$ is maxmimizing reward as written in constraints, what are we maxmizing in the objective?
  - On line 194, it is written: $\max E _ {\mathbf{s} \sim \rho_\pi}[-\log p(\mathbf{s})]$; is $p(x)$ $\rho_\pi(s)$?
  - The proof of equation (1) seems to be wrong. in equation (4), it is argued that $\rho_\pi \le p^*$ in $\mathcal{S}_2$, but in that case, the inequality should be in the opposite direction.
  - From the first place, how $ \max E_{ \mathbf{s} \sim \rho_\pi } \left[ \mathcal{H}_\pi [ \mathbf{s} ] \right] $ can be equivalent to KL minimization between $ \rho _ \pi $ and $ p ^ * $? the former converges to maxent distribution (uniform-like), whereas the latter converges to $ p ^ * $.

- While the paper defines critic conditioned state entropy and use it as its key concept, it is very hard to understand intuitively what it is, and the paper does not explain it well.
  - according to the definition, it seems like, critic conditioned state entropy averages $-\log p(s|Q(s, \pi(\cdot|s)))$ over $\rho _ \pi$. What is $p(s|Q(s, \pi(\cdot|s))$ indeed? How can a given term depend on the conditioned term?

**Questions:**

- I was not able to understand the math presented in the paper, as discussed in the weaknesses section. Then, why do we need to consider the so-called critic conditioned state entropy? I agree with the idea that we need exploration for offline-online fine-tuning, and state marginal matching can be one of exploration methods. But for that, we can simply put reward for entropy maximization; how QCSE and VCSE differ from it?
- In the experiments, QCSE adopted algorithms does improve from the baselines, but their performance is still highly correlated to the baselines' performance (e.g., walker2d -medium-replay). Why is there a gap between the theory and the experiment results? Following the papers' arguments, the proposed algorithm does state marginal matching, and shouldn't it lead to an optimal policies?

---

> ### Author Response · Authors · 2024-11-17
> **Reply to Reviewer sUP8's Weaknesses section**
>
> **We appreciate your thorough analysis of our article and the questions raised at the theoretical level. We plan to respond to your questions as soon as possible. For the parts that require revision, we will mark them in blue within the manuscript. Once we have submitted our replies to all reviewers, we will also make the necessary changes to the manuscript accordingly.**
>
> **wk.1)** In equation (1), it is said that the state entropy maximization is: $\max E_{s~\rho_{\pi}}[H(s)]$ s.t. $\pi=\arg\max[R(\tau)]$. If $\pi$ is maximizing reward as written in constraints, what are we maximizing in the objective?
>
> **re(wk.1)** In mathematics, 's.t.' is an abbreviation for 'subject to,' which signifies that it is subject to certain conditions/constraints. Therefore, the mathematical meaning of $\max E_{s \sim \rho_{\pi}}[H(s)]$ s.t. $\pi := \arg\max[R(\tau)]$ is that while maximizing entropy, we also need to find the policy that can maximize the return.
>
> In particular, The motivation for writing it in this form in the chapter (**lines 191**) is that our studying objective in this section is state entropy, but the problem we are dealing with is a RL problem.
>
> Furthermore, we distinguish the ambiguity caused by '$\max$' In the context of the QCSE setting, Q conditioned state entropy is directly added to the reward as an intrinsic reward. Therefore, maximizing 'entropy + reward' is equivalent to maximizing both the Q conditioned state entropy and accumulated Return simultaneously. During the optimization process, we are more focused on maximizing the return, so entropy can be viewed as a regularization term. However, in this chapter (**lines 191**), we aim to analyze the impact of maximizing entropy, so the RL optimization objective can be considered as a constraint.
>
> **wk.2** On line 194, it is written:$\max E_{s~\rho_{\pi}}[-\log p(s)]$, is $\rho_{\pi}$ or p(s)?
>
> **re(wk.2)** Both are acceptable. On **line 173**, we defined $\rho_{\pi}(s)$ as the marginal state distribution of $\pi$, which is expressed in the form of a density function. Therefore, we can also write it in the form of a density probability $p(s)$. Meanwhile, to eliminate any ambiguity, we consider making appropriate modifications in subsequent versions.
>
> **wk.3** The proof of equation (1) seems to be wrong. in equation (4), it is argued that $\rho_{\pi}<p^*$ in $S_2$, but in that case, the inequality should be in the opposite direction.
>
> **re (wk.3)** Thanks for pointing out this error. However, the overall conclusion will not change significantly. Specifically: We only need to swap the two domains S1 and S2 in Equation 4, i.e., changing $\max E_{s\sim\rho_{\pi}(s)}[H_{\pi}(s)]\leq\max\limits_{\rho_{\pi}(s) \atop{s.t. \max {E}_{\tau \sim \pi}[R(\tau)]}}-\int_1 \rho\cdot\log \rho-\int_2 p^*\cdot\log p^*$ to:
>
> $\max E_{s\sim\rho_{\pi}(s)}[H_{\pi}(s)]\leq\max\limits_{\rho_{\pi}(s) \atop{s.t. \max {E}_{\tau \sim \pi}[R(\tau)]}}-\int_2 \rho\cdot\log \rho-\int_1 p^*\cdot\log p^*$
>
> We can then draw the conclusion that the domain shift is mitigated. The reason is that in the $S1$ domain, there exists $\rho_{\pi}>p^*$, so $-\int_1 p^*\cdot\log p^*\geq-\int_1 \rho_{\pi}\cdot\log \rho_{\pi}$. Meanwhile, the previous analysis can be modified accordingly:
>
> 'Since $p^*(s)$ remains invariant during the training process, maximizing $J_{\rm term 2}$ is equivalent to narrowing down the domain $\textcolor{blue}{S_{1}}$. Meanwhile, maximizing $J_{\rm term 1}$ is equivalent to $\textcolor{blue}{\textit{encourage exploring $S_2$}}$. Both $J_{\rm term_1}$ and $J_{\rm term_2}$ narrow the gap between $\rho_{\pi}(s)$ and $p^*(s)$.'
>
> **wk.4** While the paper defines critic conditioned state entropy and use it as its key concept, it is very hard to understand intuitively what it is, and the paper does not explain it well.  according to the definition, it seems like, critic conditioned state entropy averages $-\log p(s|Q(s,\pi(\cdot|s)))$ over $\rho_{\pi}$. What is $p(s|Q(s,\pi(\cdot|s)))$ indeed? How can a given term depend on the conditioned term?
>
> **re (wk.4)** In section Advantages of QCSE, about **255 to 280 lines**, we analyze the limitations of VCSE by referring to an example used in this paper: Suppose there are two transitions with the same state, $T_1=(s,a_1,s_1)$ and $T_1=(s,a_2,s_2)$. If we use $V$ to compute the condition, both transitions will have the same condition. However, based $Q$ conditioning can distinguish between the two transitions. Therefore, utilizing as the $V$ condition may lead to excessive exploration of poor samples when maximizing entropy, based $Q$ conditioning can help avoid this issue. Furthermore, as illustrated in Figure 2, QCSE consistently outperforms VCSE during training, thereby validating our claim.
>
> # Reference
>
> [1] Accelerating Reinforcement Learning with Value-Conditional State Entropy Exploration.
>
> [2] Efficient Exploration via State Marginal Matching
>
> [3] State Entropy Maximization with Random Encoders for Efficient Exploration

---

> > ### Comment · Reviewer_sUP8 · 2024-11-17
> >
> > on re (wk.1). "while maximizing entropy, we also need to find the policy that can maximize the return." -> This means that among the policies that give identical return (which is maximal), we pick the policy with maximized entropy. In other words, we are taking uniform random policy over a number of optimal actions. If there is only one optimal action per state, we would be sticking to a deterministic policy.
> >
> > In QCSE setting, as you referred, you are maximizing entropy + reward. This means that you might be sacrificing reward if you can get more entropy compared to the sacrificed reward; in this case, there is no chance that we optimize to a deterministic policy, as we would get negative infinity entropy when we have deterministic policy. Thus, these two are totally different things, and we usually refer to the latter in the maximum entropy RL. This is why I asked.

---

> > > ### Comment · Reviewer_sUP8 · 2024-11-17
> > >
> > > on re (wk.4). The question was what Q-conditioned state probability itself is, not to compare it against V-conditioned probability. As far as I understood, $p(s|Q(s,a)=10)$ means the probability distribution of states with $Q$ values identical to 10. Is that correct? If it is, what does it mean to maximize its expectation? I would like to hear some intuitions about it.
> > >
> > > Furthermore, isn't $Q(s, \pi(\cdot|s))=\mathbb{E}_\pi[ Q(s,a) ]=V(s)$? I am not sure how two differs, if we refer to the definition 3 about QCSE.

---

> ### Author Response · Authors · 2024-11-17
> **Reply to Reviewer sUP8's Question section**
>
> **Q.1** I was not able to understand the math presented in the paper, as discussed in the weaknesses section. Then, why do we need to consider the so-called critic conditioned state entropy? I agree with the idea that we need exploration for offline-online fine-tuning, and state marginal matching can be one of exploration methods. But for that, we can simply put reward for entropy maximization; how QCSE and VCSE differ from it?
>
> **re (Q.1)** We answer this question by comparing different exploration methods.
>
>  - **VCSE>SE>RND** First, [2] points out that the RND-based policy struggles to converge to a single exploratory policy and addresses this issue by proposing State Marginal Matching (SMM). We analyze and suggest that maximizing state entropy can implicitly achieve SMM, indicating that QCSE/VCSE/SE are likely superior to RND. On the other hand, [3] proposes State Entropy Maximization (SE) as an intrinsic reward to encourage exploration, with experimental results showing that SE outperforms RND. Subsequently, [1] introduces VCSE to address SE's limitation of potentially over-exploring low-value samples. Generally, the results should reflect the following relationship: VCSE > SE > RND.
>
> - **QCSE>VCSE** mentioned in **wk.4**
>
> Therefore, QCSE>VCSE>SE>RND. Meanwhile, the experimental results in our Table 3 show that QCSE has better experimental performance compared to VCSE and RND when utilizing CQL and Cal-QL.
>
> **Q.2** In the experiments, QCSE adopted algorithms does improve from the baselines, but their performance is still highly correlated to the baselines' performance (e.g., walker2d -medium-replay). Why is there a gap between the theory and the experiment results? Following the papers' arguments, the proposed algorithm does state marginal matching, and shouldn't it lead to an optimal policies?
>
> **re (Q.2)** Thanks for your question. We acknowledge that the reviewer has high-quality expectations for theoretical details. We did not claim that this is the optimal algorithm in human history. The term "optimal" is mentioned in many articles, but new algorithms always emerge to achieve better performance. If the reviewer has concerns about the mathematical description of this term, we consider using a more appropriate word such as "better performance" to replace "optimal."
>
> Secondly, our paper conducts theoretical analysis based on soft-Q, and in the corresponding sections of the article, we also state that our algorithm theoretically guarantees improvements for soft-Q-based algorithms (**Theorem (Converged QCSE Soft Policy is Optimal)**, lines 953~955). Meanwhile, if an algorithm can ensure policy improvement, it will asymptotically converge to the optimal policy through iterations. Based on the theoretical analysis of QCSE on soft-Q-based algorithms, in Figure 3, we mainly select various soft-Q-based algorithms for testing and achieve better performance compared to the baseline algorithms. Additionally, we have not conducted theoretical analysis on algorithms other than soft-Q. Therefore, in Figure 4, we test algorithms beyond soft-Q, and the experimental results show that QCSE can improve these algorithms as well.
>
> # Reference
> [1] Accelerating Reinforcement Learning with Value-Conditional State Entropy Exploration.
>
> [2] Efficient Exploration via State Marginal Matching
>
> [3] State Entropy Maximization with Random Encoders for Efficient Exploration

---

> ### Author Response · Authors · 2024-11-17
> **Further reply to the Official Comment by Reviewer sUP8**
>
> Thank you very much for further analyzing the details. We think your suggestion makes sense. Therefore, we understand that your main concern is that we should not analyze the "return" and "entropy" separately in the context of "entropy + return". As you mentioned, in maximum entropy RL, what we maximize is not the return but the soft-Q value.
>
> To our knowledge, algorithms based on soft-Q values belong to a category of algorithms in max entropy RL, such as Soft Actor-Critic (SAC), and some inverse RL algorithms are also based on soft-Q values. Therefore, the theoretical analysis presented in the main text (**Theorem (Converged QCSE Soft Policy is Optimal)**) still falls within this category.
>
> However, when analyzing entropy, we cannot ignore the fact that we are optimizing within the context of max state entropy RL algorithms. Therefore, we consider replacing the "s.t." and subsequent "$\max R(\tau)$" with a verbal description, such as "in the context of maximum entropy RL," because we cannot analyze entropy in isolation from RL problems. Alternatively, we will seek more appropriate conditions for further substitution. Thank you.

---

> ### Author Response · Authors · 2024-11-17
> **Further reply to the Official Comment by Reviewer sUP8**
>
> **Q.1 [wk.4 re]** on re (wk.4). The question was what Q-conditioned state probability itself is, not to compare it against V-conditioned probability. As far as I understood,
>  means the probability distribution of states with
>  values identical to 10. Is that correct? If it is, what does it mean to maximize its expectation? I would like to hear some intuitions about it.
>
> **re (Q.1  [wk.4 re])** If we step back and consider this problem, when the condition is V, this probability is state-dependent. When we use Q as the condition, this probability is state action-dependent. When maximizing the $-\log$ probability of the V condition, state samples with the same value will be adequately explored. Correspondingly, when maximizing the probability of the Q condition, state samples with the same Q value will be explored.
>
> The advantage of this approach lies in the ability to further distinguish samples because V is the expectation of Q. Having the same V value does not necessarily mean that all RL samples $(s, a, r)$ are equivalent. This can reduce the exploration of states in high-value $(s, a, r)$ samples while increasing the exploration of states in low-value tuples. Furthermore, using Q values as conditions allows for further distinction of samples that cannot be differentiated when using V values as conditions, thereby further ensuring the **stability** of online fine-tuning.
>
> **Q.2 [wk.4 re]** Furthermore, isn't ? I am not sure how two differs, if we refer to the definition 3 about QCSE.
>
> **re (Q.2 [wk.4 re])** Since $E_{\pi}[Q(s,a)]=\int_{a\sim\pi(\cdot|s)} p(a|s) Q(s,a) da=V(s)$, it follows that  $Q(s,a)!=V(s)=E_{\pi}[Q(s,a)]]$.
>
> Meanwhile, the motivation we write $Q(s,\pi(\cdot|s))$ instead of $Q(s,a)$ is that we didn’t utilize replay buffer this notation in this section, if we have to utilize this notation, we must have to define offline and online replay buffers to help to denote $a\sim D_{online}$.
>
> Therefore, for convenience, we write $Q(s,\pi(\cdot|s))$ rather $Q(s,a)$. note: $a:=\pi(\cdot|s)$

---

> ### Author Response · Authors · 2024-11-30
> **Official comments by Authors**
>
> Dear reviewer
>
> We kindly ask you to further confirm whether we have addressed your concerns.
>
> Thank you.

---

### Official Review · Reviewer_L7Sj · 2024-11-03

**Soundness:** 3
**Presentation:** 2
**Contribution:** 3
**Rating:** 5
**Confidence:** 3

**Summary:**

This paper explores strategies for optimizing the fine-tuning of pre-trained policies in offline reinforcement learning (RL) to enhance sample efficiency. Traditional fine-tuning methods often fall short due to a distribution mismatch between offline pre-training and subsequent online fine-tuning, which restricts the quality of online sample acquisition and hampers performance. To mitigate this distribution shift, the authors introduce Q-conditioned state entropy (QCSE) as an intrinsic reward mechanism. QCSE maximizes the entropy of individual states based on their respective Q-values, promoting exploration of underrepresented samples while disincentivizing overrepresented ones. This approach implicitly aligns with State Marginal Matching (SMM), resolving the asymptotic limitations often seen in constraint-based techniques. Moreover, QCSE integrates seamlessly with various RL algorithms, leading to significant improvements in online fine-tuning. Experimental results show approximately 13% performance gains for CQL and 8% for Cal-QL, with additional tests confirming QCSE’s versatility across different algorithms.

**Strengths:**

1. The idea proposed in this paper has a degree of novelty, as it connects the offline-to-online learning problem with exploration mechanisms, leading to a new algorithm. This represents a relatively fresh perspective.

2. The experiments in this paper are quite comprehensive, evaluating the algorithm's effectiveness, modularity, and advancement from multiple perspectives. Additionally, the experiments cover a wide range of tasks, providing robust empirical support for the proposed method.

**Weaknesses:**

My main concerns about this paper lie in the method description section, where I feel that several key points are not fully explained by the authors.

1. Regarding the interpretation of Equation (1), if the goal is to maximize the right-hand side $J_2$, since the optimal policy’s corresponding $p^*$ is fixed, it’s clear that the scope of $S_2$ should be expanded, rather than "narrowing down the domain $S_2$" as the authors suggest. Meanwhile, to maximize $J_1$, it’s evident that the range of $S_1$ should be reduced, i.e., $p(s)$ over $S_1$ should be decreased to approach $p^*(s)$. Therefore, the overall approach seems to be "increasing exploration in $S_2$ while penalizing exploration in $S_1$." However, this is contrary to the authors’ description in lines 203–209. Could you clarify your reasoning behind the claim of "narrowing down the domain $S_2$" and explain how this aligns with maximizing $J_2$. Additionally, you could  revisit the description in lines 203-209 to ensure consistency with the mathematical formulation.

2. In the "Implementation of QCSE" section, I did not see how the practical algorithm for QCSE was derived from Equation (1). Specifically, what do the $S_1$ and $S_2$ in Equation (1) correspond to here? What role does maximizing the Critic Conditioned State Entropy $H(s|Q)$ in Equation (2) play, and how is it related to Equation (1)? Particularly, if there are indeed issues with the interpretation of Equation (1) mentioned above, does that mean Equation (2) would not hold either? The authors are suggested to provide a step-by-step derivation showing how the practical algorithm in Equation (2) follows from the theoretical formulation in Equation (1). Specifically, could you clarify on how $S_1$ and $S_2$ from Equation (1) are represented in the implementation, and how maximizing $H(s|Q)$ relates to the objectives in Equation (1)?

3. The origin of Equation (3) is also unclear. The authors simply refer to the literature, but they do not explain how (3) is derived from (2), which is not obvious. Therefore, the authors have an obligation to provide the derivation process from (2) to (3).

**Questions:**

The same as Weaknesses.

---

> ### Author Response · Authors · 2024-11-18
> **Reply to Reviewer L7Sj's Weaknesses section (part 1)**
>
> **wk.1** Regarding the interpretation of Equation (1), if the goal is to maximize the right-hand side $J_2$, since the optimal policy’s corresponding $p^*$ is fixed, it’s clear that the scope of $S_2$ should be expanded, rather than "narrowing down the domain $S_2$" as the authors suggest. Meanwhile, to maximize $J_1$, it’s evident that the range of $S_1$ should be reduced, i.e., $p(s)$ over $S_1$ should be decreased to approach . Therefore, the overall approach seems to be "increasing exploration in $S_2$ while penalizing exploration in $S_1$ ." However, this is contrary to the authors’ description in lines 203–209. Could you clarify your reasoning behind the claim of "narrowing down the domain $S_2$" and explain how this aligns with maximizing $J_2$. Additionally, you could revisit the description in lines 203-209 to ensure consistency with the mathematical formulation.
>
> **re (wk.1)** Thanks for this question. This issue arises from an error in our proof of Equation (1), where we incorrectly stated that in domain $S_2$ it have $-p^*\log p^*<-\rho\log \rho$, since $p^*\log p^*<\rho\log \rho$. Consequently, the direction of the inequality in Equation (1) is incorrect, and a similar issue was also pointed out by the reviewer sUP8.
>
> However, by swapping the two domains $S_1$ and $S_2$, we can still derive that optimizing Equation 1 is equivalent to reducing the distribution shift between the offline and online stages.
>
> Specifically, we only need to swap the two domains $S_1$ and $S_2$ in Equation (4), i.e., changing $\max E_{s\sim\rho_{\pi}(s)}[H_{\pi}(s)]\leq\max\limits_{\rho_{\pi}(s) \atop{s.t. \max {E}_{\tau \sim \pi}[R(\tau)]}}-\int_1 \rho\cdot\log \rho-\int_2 p^*\cdot\log p^*$ to:
>
> $\max E_{s\sim\rho_{\pi}(s)}[H_{\pi}(s)]\leq\max\limits_{\rho_{\pi}(s) \atop{s.t. \max {E}_{\tau \sim \pi}[R(\tau)]}}-\int_2 \rho\cdot\log \rho-\int_1 p^*\cdot\log p^*$
>
> We can then draw the conclusion that the domain shift is mitigated. The reason is that in the $S1$ domain, there exists $\rho_{\pi}>p^*$, so $-\int_1 p^*\cdot\log p^*\geq-\int_1 \rho_{\pi}\cdot\log \rho_{\pi}$. Meanwhile, the previous analysis can be modified accordingly:
>
> 'Since $p^*(s)$ remains invariant during the training process, maximizing $J_{\rm term 2}$ is equivalent to narrowing down the domain $\textcolor{blue}{S_{1}}$. Meanwhile, maximizing $J_{\rm term 1}$ is equivalent to $\textcolor{blue}{\textit{encourage exploring $S_2$}}$. Both $J_{\rm term_1}$ and $J_{\rm term_2}$ narrow the gap between $\rho_{\pi}(s)$ and $p^*(s)$.'
>
> **wk.2** In the "Implementation of QCSE" section, I did not see how the practical algorithm for QCSE was derived from Equation (1). Specifically, what do the $S_1$ and $S_2$ in Equation (1) correspond to here? What role does maximizing the Critic Conditioned State Entropy $H(s|Q)$ in Equation (2) play, and how is it related to Equation (1)? Particularly, if there are indeed issues with the interpretation of Equation (1) mentioned above, does that mean Equation (2) would not hold either? The authors are suggested to provide a step-by-step derivation showing how the practical algorithm in Equation (2) follows from the theoretical formulation in Equation (1). Specifically, could you clarify on how $S_1$ and $S_2$ from Equation (1) are represented in the implementation, and how maximizing $H(s|Q)$ relates to the objectives in Equation (1)?
>
> **re (wk.2)** Thanks for your questions. $S_1$ and $S_2$ are subsets of the entire state sample space, i.e., $S = S_1 \cup S_2$. Meanwhile, in the $S_1$ domain, there exists $\rho_{\pi}>p^*$, while there exists $\rho_{\pi}<p^*$ in the $S_2$ domain (lines 199 to 202).
>
> When responding to reviewer wk.1, we pointed out the issues(we should swap $S_1$ and $S_2$) with Equation (1), but these do not affect the conclusion that maximizing entropy can help to mitigate the distribution shift from the offline to the online stage. The role of Equation 1 is to provide an intuitive understanding of the relationship between maximizing state entropy and reducing the offline-to-online distribution shift.
>
> The connection between Equation 2 and Equation 1 is that Equation 2 represents a form of intrinsic reward based on Q-conditioned state entropy. This intrinsic reward, similar to state entropy as an intrinsic reward, serves to encourage exploration by the agent. However, Q-conditioned state entropy, akin to VCSE, can explore samples with different Q values separately, thus ensuring the stability of training. We have discussed above that maximizing state entropy can help reduce the distribution shift between offline and online settings. Therefore, Equation 1 serves to illustrate the effectiveness of Equation 2.

---

> ### Author Response · Authors · 2024-11-18
> **Reply to Reviewer L7Sj's Weaknesses section (part 2)**
>
> **wk.3** The origin of Equation (3) is also unclear. The authors simply refer to the literature, but they do not explain how (3) is derived from (2), which is not obvious. Therefore, the authors have an obligation to provide the derivation process from (2) to (3).
>
> **re (wk.3)** We explain the Eq (3) and how it relates to maximize entropy.
>
> Given $n$ capacity samples $X=[x_1=(a_1,b_1),\cdot,x_n=(a_n,b_n)]$, we define the L2 distance between $x_i$ and $x_j$ as $L_2(x_i,x_j)=||a_i-a_j||_2+||b_i-b_j||_2$. The expression $P(L(x_i,x_j)\leq\delta|(a_j,b_j)\in X, i!=j)=\frac{N(X')}{X}$ quantifies the percentage of samples within a distance of $\delta$ from $(a_i,b_i)$, where $X'$ denotes all $x_i$ that satisfy $L_2(x_i,x_j)\leq\delta$, and $(a_j,b_j)\in X$, $i!=j$. Moreover, if $\delta$ is a very small number, $P(L(x_i,x_j)\leq\delta|(a_j,b_j)\in X, i!=j)=\frac{N(X')}{X}$ can be used to estimate the density probability of $x_i$.
>
> Subsquently, by maximizing $\sum_{x_j}L_2(x_i,x_j)$, fewer samples $x_j$ in the set $X$ can be within a distance of $\delta$ from $x_i$, thereby reducing the density probability of $x_i$.
>
> Next, we discuss the relationship between Eq (3) and the maximization of $H(X)$. Since $H(X)=\sum_i -\log P(x_i)$, minimizing $H(X)=\sum_i P(x_i)$ is equivalent to maximizing $H(X)$. Simultaneously, maximizing the second term in Eq (3) is approximately equivalent to maximizing $\sum_{x_j\in X}L(x_i,x_j)$. (Note: In Eq (3), $x_j$ refers only to the knn of $x_i$.) Therefore, maximizing Eq (3) is equivalent to minimizing $P(L(x_i,x_j)\leq\delta|(a_j,b_j)\in X, i!=j)=\frac{N(X')}{X}$, which in turn maximizes $H(X)$.

---

> ### Author Response · Authors · 2024-11-30
> **Official comments by Authors**
>
> Dear reviewer
>
> We kindly ask you to further confirm whether we have addressed your concerns.
>
> Thank you.

---

> > ### Comment · Reviewer_L7Sj · 2024-12-02
> >
> > Thank you very much for your response! After thoroughly considering the feedback from other reviewers and your comments, I have decided to maintain my score.

---

> > > ### Author Response · Authors · 2024-12-02
> > > **Reply to Reviewer L7Sj**
> > >
> > > Thank you for raising the critical issues. The corresponding revisions in the paper have been marked in blue in the main text.

---

### Official Review · Reviewer_JySu · 2024-11-05

**Soundness:** 3
**Presentation:** 3
**Contribution:** 2
**Rating:** 6
**Confidence:** 3

**Summary:**

This paper proposes a new reinforcement learning method called Q-conditioned State Entropy Maximization (QCSE) which aims to improve the performance of offline-to-online RL process. The authors prove that QCSE can achieve State Marginal Matching (SMM), an exploration strategy theoretically ensuring optimal performance. Experiments show that QCSE significantly enhances the performance of existing model-free algorithms like CQL and Cal-QL, with an average improvement of about 13% and 8%. Additionally, QCSE exhibits general applicability to other model-free algorithms such as SAC, IQL, and TD3+BC.

**Strengths:**

This paper presents a novel approach to offline-to-online reinforcement learning, demonstrating strong originality and significance. The authors provide clear and thorough explanations of their proposed method, making the paper highly readable and understandable.

**Weaknesses:**

On the novelty factor, contributions are not very significant.

The paper would benefit from a more thorough analysis and discussion of the limitations and potential challenges associated with the proposed framework.

The experimental evaluation should include comparisons with a broader range of state-of-the-art methods to offer a more comprehensive assessment of QCSE's performance.

The theoretical analysis is insufficient and needs strengthening.

**Questions:**

Could you please provide more clarity on how the hyper-parameter settings, particularly the choice of  λ and the number of k-nearest neighbor (knn) clusters, affect the results? I would appreciate a more detailed explanation of their impact on the performance of QCSE, as the current description is a bit unclear to me.

---

> ### Author Response · Authors · 2024-11-19
> **Reply to Reviewer L7Sj's question section**
>
> **Q.1.1** Could you please provide more clarity on how the hyper-parameter settings, particularly the choice of λ and the number of k-nearest neighbor (knn) clusters, affect the results? **Q.1.2** I would appreciate a more detailed explanation of their impact on the performance of QCSE, as the current description is a bit unclear to me.
>
> **re (Q.1.1)** We first brifly introduce the role of λ and knn.
>
> The role of λ is primarily to balance the importance between intrinsic rewards and actual rewards. A larger λ means that during the process of optimizing the policy, the speed of maximizing entropy will be faster, which will increase the exploration of the policy, but it may also affect the stability of training. This may be because a certain degree of stability is required during the online fine-tuning phase, and excessive exploration can affect the effectiveness of the policy's fine-tuning. During our testing of QCSE, we did not make extensive adjustments to λ. Instead, we found that setting λ to a fixed value of 1 and adjusting knn could achieve better results.
>
> Similarly to λ, the choice of knn can also affect the performance of the algorithm. The choice of knn has an impact on the experimental performance of QCSE, and its effectiveness can be illustrated from the perspective of ablation experiments, complemented by Figure 2. We further detailed the mechaism in **re (Q.1.2)**
>
> **re (Q.1.2)** First, we introduce the relationship between the KGS estimator in Equation 3 and maximizing entropy. Then, we explain why the choice of knn affects the maximization of state entropy.
>
> **KSG estimator and state entropy maximization.** Given $n$ capacity samples $X=[x_1=(a_1,b_1),\cdot,x_n=(a_n,b_n)]$, we define the L2 distance between $x_i$ and $x_j$ as $L_2(x_i,x_j)=||a_i-a_j||_2+||b_i-b_j||_2$. The expression $P(L(x_i,x_j)\leq\delta|(a_j,b_j)\in X, i!=j)=\frac{N(X')}{X}$ quantifies the percentage of samples within a distance of $\delta$ from $(a_i,b_i)$, where $X'$ denotes all $x_i$ that satisfy $L_2(x_i,x_j)\leq\delta$, and $(a_j,b_j)\in X$, $i!=j$. Moreover, if $\delta$ is a very small number, $P(L(x_i,x_j)\leq\delta|(a_j,b_j)\in X, i!=j)=\frac{N(X')}{X}$ can be used to estimate the density probability of $x_i$.
>
> Subsquently, by maximizing $\sum_{x_j}L_2(x_i,x_j)$, fewer samples $x_j$ in the set $X$ can be within a distance of $\delta$ from $x_i$, thereby reducing the density probability of $x_i$.
>
> Next, we discuss the relationship between Eq (3) and the maximization of $H(X)$. Since $H(X)=\sum_i -\log P(x_i)$, minimizing $H(X)=\sum_i P(x_i)$ is equivalent to maximizing $H(X)$. Simultaneously, maximizing the second term in Eq (3) is approximately equivalent to maximizing $\sum_{x_j\in X}L(x_i,x_j)$. (Note: In Eq (3), $x_j$ refers only to the knn of $x_i$.) Therefore, maximizing Eq (3) is equivalent to minimizing $P(L(x_i,x_j)\leq\delta|(a_j,b_j)\in X, i!=j)=\frac{N(X')}{X}$, which in turn maximizes $H(X)$.
>
> **knn and state entropy maximization.** When we choose a larger knn, it means that each $x_i$ will correspond to more $x_j$ during each update, thereby accelerating the process of maximizing entropy. Therefore, a larger knn encourages the agent to explore more, which has some similarities to choosing a larger λ.
>
> In summary, both λ and knn affect the importance of entropy in each update, which in turn influences the agent's exploration and further impacts the effectiveness of online fine-tuning.

---

> ### Author Response · Authors · 2024-11-30
> **Official comments by Authors**
>
> Dear reviewer
>
> We kindly ask you to further confirm whether we have addressed your concerns.
>
> Thank you.

---

### Author Response · Authors · 2024-11-23
**Paper Modifications**

We are grateful to the reviewers for their valuable questions. We have made corresponding revisions to the paper based on the reviewers' suggestions, and the changes have been marked in blue. Thank you.

- **lines (191 to 208) and lines (892 to 900)** Both reviewers sUP8 and L7Sj raised concerns about the analysis of Equation (1) and lines 191-208. In our response to sUP8, we pointed out that flipping the $S_1$ and $S_2$ domains in Equation (1) and making corresponding adjustments to the analysis below the equation would not affect the conclusion that optimizing Equation (1) facilitates reducing the offline-to-online domain shift. Additionally, we have made the necessary modifications in the corresponding sections of the paper.

- **lines (191 to 208) and lines (892 to 900)** Reviewer sUP8 pointed out that the constraint in line 194 should not be max Return, as this would imply maximum entropy RL. We agree with Reviewer sUP8 and have replaced max R with max SoftQ.

- **lines (1243~1263)** Reviewer Piga suggested that we add tests related to the medium-expert dataset. In the supplementary page F.1, we have provided the comparison of the performance differences between CQL-QCSE and CQL on the medium-expert dataset. The experimental results indicate that QCSE does not significantly enhance CQL's performance on the medium-expert dataset, which proves that the effectiveness of QCSE has already converged on the medium dataset. This further demonstrates the effectiveness of QCSE in the offline-to-online setting.

---

### Meta-Review · Area_Chair_bTiZ · 2024-12-22

**Metareview:**

The paper proposes a novel intrinsic motivation quantity for exploration in offline-to-online RL. While this is an important topic, and the paper is mostly well written and motivated, some concerns were raised regarding the mathematical clarity, and indeed correctness, of the proposed approach. Other concerns included the discussion of novelty and limitation, as well as the strength of the empirical evaluation, in particular with regard to a broader range of baselines.

**Additional Comments On Reviewer Discussion:**

Most reviewers provided thorough and reasoned concerns (one exception is the insistence of Reviewer Piga on comparison to an existing method with limited relevance). While the authors' feedback to address these are appreciated, these concerns ultimately remain.

---

### Decision · Program_Chairs · 2025-01-22

Reject